# Microplastics in the Ecosystem: An Overview on Detection, Removal, Toxicity Assessment, and Control Release

**Bhamini Pandey [1], Jigyasa Pathak [1], Poonam Singh [1,\*], Ravinder Kumar [2], Amit Kumar [3,\*], Sandeep Kaushik [4] and Tarun Kumar Thakur [4]**

1 Department of Applied Chemistry, Delhi Technological University, Delhi 110042, India
2 Department of Chemistry, Gurukula Kangri (Deemed to be University), Haridwar 249404, Uttarakhand, India
3 School of Hydrology and Water Resources, Nanjing University of Information Science and Technology, Nanjing 210044, China
4 Department of Environmental Science, Indira Gandhi National Tribal University, Amarkantak 484887, Madhya Pradesh, India
\* Correspondence: poonam@dtu.ac.in (P.S.); amitkdah@nuist.edu.cn (A.K.)

**Abstract:** In recent decades, the accumulation and fragmentation of plastics on the surface of the planet have caused several long-term climatic and health risks. Plastic materials, specifically microplastics (MPs; sizes < 5 mm), have gained significant interest in the global scientific fraternity due to their bioaccumulation, non-biodegradability, and ecotoxicological effects on living organisms. This study explains how microplastics are generated, transported, and disposed of in the environment based on their sources and physicochemical properties. Additionally, the study also examines the impact of COVID-19 on global plastic waste production. The physical and chemical techniques such as SEM-EDX, PLM, FTIR, Raman, TG-DSC, and GC-MS that are employed for the quantification and identification of MPs are discussed. This paper provides insight into conventional and advanced methods applied for microplastic removal from aquatic systems. The finding of this review helps to gain a deeper understanding of research on the toxicity of microplastics on humans, aquatic organisms, and soil ecosystems. Further, the efforts and measures that have been enforced globally to combat MP waste have been highlighted and need to be explored to reduce its potential risk in the future.

**Keywords:** microplastics; environmental pollution; COVID-19; detection techniques; toxicity assessment

## 1. Introduction

Today's world relies heavily on plastic on a global scale, infiltrating almost every aspect of human lives. Plastics are organic polymers that exhibit exceptional properties such as durability, flexibility, lightness, and mechanical and thermal stability which contribute to their widespread applications in construction, food and packaging industries, pharmaceuticals, and many more sectors [1]. Despite annual expansion in the plastic industry, the demand for plastic does not seem to be decreasing. The amount of plastic generation is estimated to reach approximately 33 billion tons by the year 2050 [2]. The environmental impact of plastic has been a considerable concern for government entities, the scientific community, and the general public, regardless of its long-term industrial benefit [3]. The production and distribution of plastics possessing high degradation resistance are increasing at a rapid pace, which has serious environmental and ecological consequences. Geyer et al. [4] reported the contamination of the marine environment by 4–12 million metric tons of land-generated plastic waste by 2010.

Environmental pollution caused by plastic debris has become increasingly apparent in the past few decades. Although the size of plastic debris can range from microscopic particles to pieces measuring several meters in length, the focus of public concern is currently on synthetic microplastics having a diameter of less than 5 mm [5]. The term microplastics





(MPs) was first given by Thompson et al. [6]. Table 1 presents a number of frequently used commodity plastics along with their structure, applications, and associated hazards. For instance, polypropylene is a commonly used commodity plastic that may naturally degrade in approximately 30 years, potentially causing an unknown harmful impact on the biosphere [7]. In addition, these MPs also serve as carriers of various hazardous pollutants in biomedical and cosmetic products as well as some organic contaminants such as polychlorinated biphenyls (PCBs), dichlorodiphenyltrichloroethane (DDT), polycyclic aromatic hydrocarbons (PAHs), etc. [8]. The increasing contribution of microplastics to the environment has resulted in MP pollution becoming a global issue.

**Table 1.** Applications and hazards associated with commodity microplastics.

| Polymer | Structure | Applications | Toxic Effects | References |
|---|---|---|---|---|
| Polyethylene (PE) |  | Packaging | Detrimental to environment | [9] |
| Polyethylene terephthalate (PET) |  | Packaging | Disruption of endocrine system | [10] |
| Polypropylene (PP) |  | Automotives and furniture | Carcinogenic and cytotoxic | [11] |
| Polystyrene (PS) |  | Food packaging | Inhibition of growth and mortality | [12] |
| Polyvinyl chloride (PVC) |  | Constructions and buildings | Damage to immune system and causes infertility | [13] |
| Polyurethane (PU) |  | Constructions and buildings | Cause neurological impairment | [14] |
| Polyamide (PA) |  | Textiles and automotives | Liver damage | [15] |

### 1.1. Properties of MPs

A vast array of products made from plastic are used in day-to-day life, including packaging, containers, coatings, bags, etc. Since microplastics are chemically stable, they can last for thousands of years or longer in the environment [16]. Around 90% of the total plastics produced in the world are polymeric materials, including PET, PS, PVC, and PE [17]. The physicochemical properties of these polymers determine how these microsized particles interact under different environmental conditions. The interaction between MPs and biota depends upon the size of the plastics. Three different sizes of microsized PS beads were examined to determine their impact on the survival and development of marine copepods *T. japonica* [18]. Their findings revealed that the MPs of PS may have detrimental effects on marine copepods, including decreased survival and retarded development [18]. Another important property determining the interaction between MPs and biological systems is particle shape. Au et al. [19] estimated the impacts of the shape of polypropylene MPs (beads and fibers) on the development, reproduction, and egestion of amphipod *Hyalella azteca*. Compared to beads, MP fibers exhibited more toxicity owing to their prolonged residence time in the gut, causing food to be egested more slowly and the growth of amphipods to be significantly slower [19]. The irregularities in the shape of MPs result in their more rapid attachment to the external and internal surfaces of the terrestrial or aquatic biota.

Several chemical characteristics determine the chemical nature of microplastics, including functional groups, surface polarities, stability, and crystallinity. The chemical properties of MPs are associated with their affinity for chemicals, as opposed to their physical properties that directly affect ingestion, egestion, or cause physical injury to marine and terrestrial biota. Microplastics tend to accumulate other pollutants from the ecosystem primarily due to their polarities and functional groups. The adsorption of 18 perfluoroalkyl (PFAS) compounds on three different MPs (PS, PE), and carboxylate polystyrene (PS-COOH) was studied by Llorca et al. [20], where it was observed that PS and PS-COOH exhibited higher affinity for PFASs than PE. In addition, it was also concluded that the interactions between PFASs and microplastics increased the toxicity of hydrophobic contaminants in terrestrial and aquatic ecosystems. The crystallinity of a polymer determines the physicochemical properties of MPs, i.e., permeability, density, etc., which successively govern their hydration and swelling properties. Chen et al. [21] illustrated that the degree of crystallinity of MPs alters with degradation time. They studied the biodegradation of polycaprolactone (PCL) MPs and observed that MPs became more crystalline upon degradation, demonstrating a preference for degradation in the amorphous region. This may lead to the formation of crystallites exhibiting different toxicity in comparison to the parent microplastic materials [21].

### 1.2. Primary and Secondary MPs

There are a variety of routes through which microplastics can enter the environment, and they can be classified as primary or secondary MPs, based on their origin and usage. Primary MPs are materials purposely fabricated for a particular application; however, secondary MPs are formed as a byproduct of the fragmentation and breakdown of larger microplastics via hydrolysis, UV rays, mechanical friction, etc. [22]. The common sources of both primary and secondary MPs are given in Figure 1.

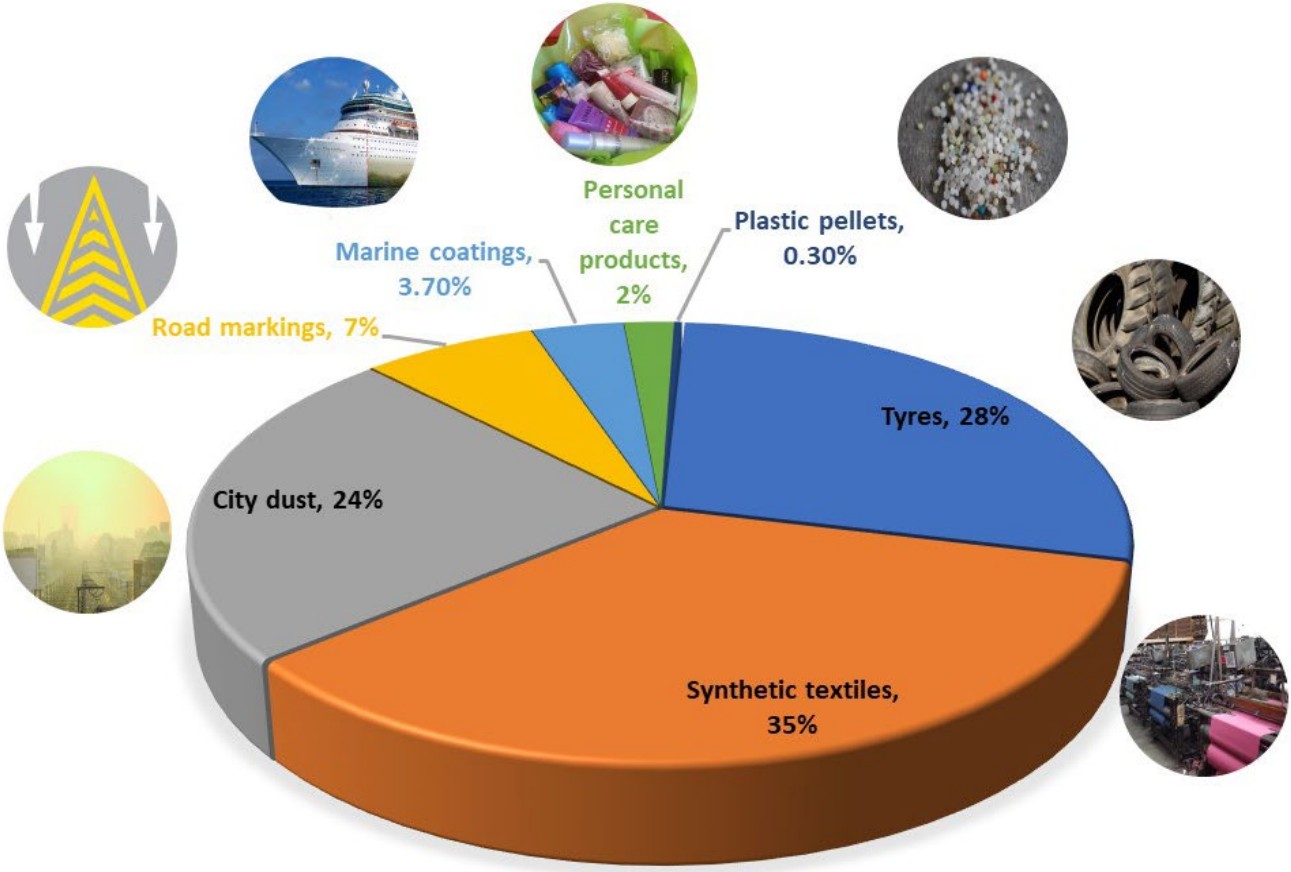

**Figure 1.** Sources of primary and secondary MPs [23].

Generally, primary microplastic materials come from various sources, including plastic pellets, vectors for drugs, and cosmetics products. It is still unclear where and how primary microplastics are produced, especially concerning the amounts of each type of microplastic that are released. Plastic pellets and flakes, used in making plastic products, are among the primary sources of microplastics [24]. These MPs can be released into the environment through an accidental loss during transportation or contamination during processing if they are not handled properly [25]. Certain segments of personal care products, including hand cleaners, sanitizers, facial cleansers, sunscreen, and toothpaste, use microplastic particles as exfoliants. The market has gradually been replaced by products containing microbeads instead of natural materials such as pumice, apricots, walnut peel, etc. [26,27]. According to a survey conducted by Cosmetics Europe (2012) in Switzerland, European Union, and Norway, polyethylene accounted for 93% of MPs employed in skin care products [28]. The usage of MPs in medical applications, such as tooth polish for dentistry, and pharmaceutical carriers, is also widespread. These MPs from cosmetic and medical products are released into the natural environment after usage, leading to aggravation of MPs pollution.

Fragmentation of plastic materials triggers the release of secondary MPs in the ecosystem, which occurs when plastic is degraded into smaller pieces as a result of various processes. Plastic waste enters the nearby water bodies because of littering and improper waste handling. Various natural weathering processes such as UV irradiation, pH changes, biological activities, exposure to particular chemicals, etc., result in the fragmentation or degradation of plastic waste into secondary MPs (Figure 2) [16,29,30].

# Weathering/aging processes of Microplastics

**Figure 2.** Weathering or ageing processes of microplastics.

Fishing gears, fish cages, and fishnets are also sources of secondary MPs, although these items are not intended to release microplastic particles into water resources, they do so when they deteriorate over time. Among fishing waste, nylon nets and fibrous ropes are the most commonly lost wastes during fishing activities [24]. It has been identified that washing synthetic clothes is also considered a significant source of MP contamination in the environment. According to a study by Napper and Thompson [31], in garment industry, each cloth product releases 1900 MP particles into the wastewater during the washing process [25]. Automobile tire abrasion has also been considered another source of MPs in the environment. As vehicles are driven on roads, the elastomers on their tires wear out and abrasion of tires occurs, which results in fine dust pollution as well as MP pollution. Approximately 0.81 Kg of abrasion from tires gets into the environment annually [26].

## 1.3. Impacts of COVID-19 on Release of MPs in the Environment

Pandemic-threatening contagious diseases have emerged and spread across history regularly. Humankind has already suffered from various pandemics and epidemics, including plague, cholera, famine, and Middle East respiratory syndrome coronavirus (MERS-CoV) [32,33]. A global pandemic has been sweeping the world since December 2019 caused by the novel coronavirus (SARS-CoV-2) suspected of causing a severe respiratory illness, termed COVID-19. The WHO (World Health Organization) declared COVID-19 a global pandemic in March 2020, and since then, preventive measures have been adopted to control its spread. The excessive usage of single-use plastic (SUP) materials such as face masks, disposable utensils, personal protective equipment (PPE), food packaging plastics, etc., during COVID-19 led to MP discharge into the environment [34]. Therefore, a sudden increase in plastic pollution can be observed through a significant amount of generated biowaste and medical waste during the COVID-19 pandemic. In the aftermath of the COVID-19 outbreak, the global demand for personal protective equipment increased significantly, with 65 and 129 billion pairs of gloves and masks consumed each month, respectively [35]. According to Benson et al., global plastic waste has increased by 1.6 million tonnes since the beginning of the pandemic. Every day approximately 3.4 billion single-use face masks and shields are discarded [36]. In addition, according to Peng et al. (2021), as of 23 August 2021, the plastic waste generated during the pandemic by 193 countries reached over eight million tons and washed into the ocean globally more than 25 thousand tons

which is approximately 1.5% of total plastic discharged into the aquatic environment [37]. The common sources, environmental processing of generated waste, and the fate of MPs generated during COVID-19 are given in Figure 3. Morgana et al. [38] experimented to confirm the release of MPs into water from face masks made up of polypropylene (PP). They carried out the fragmentation and deterioration of three-layered surgical masks in water via a rotating blender in order to mimic the circular waves and motions of water in oceans. The experimental outcomes showed that an enormous quantity of microplastics can be discharged from a singular mask under weathering conditions. Additionally, they also reported that as the exposure time and shear intensity increased, the release of MPs from disposable masks increased too [38].

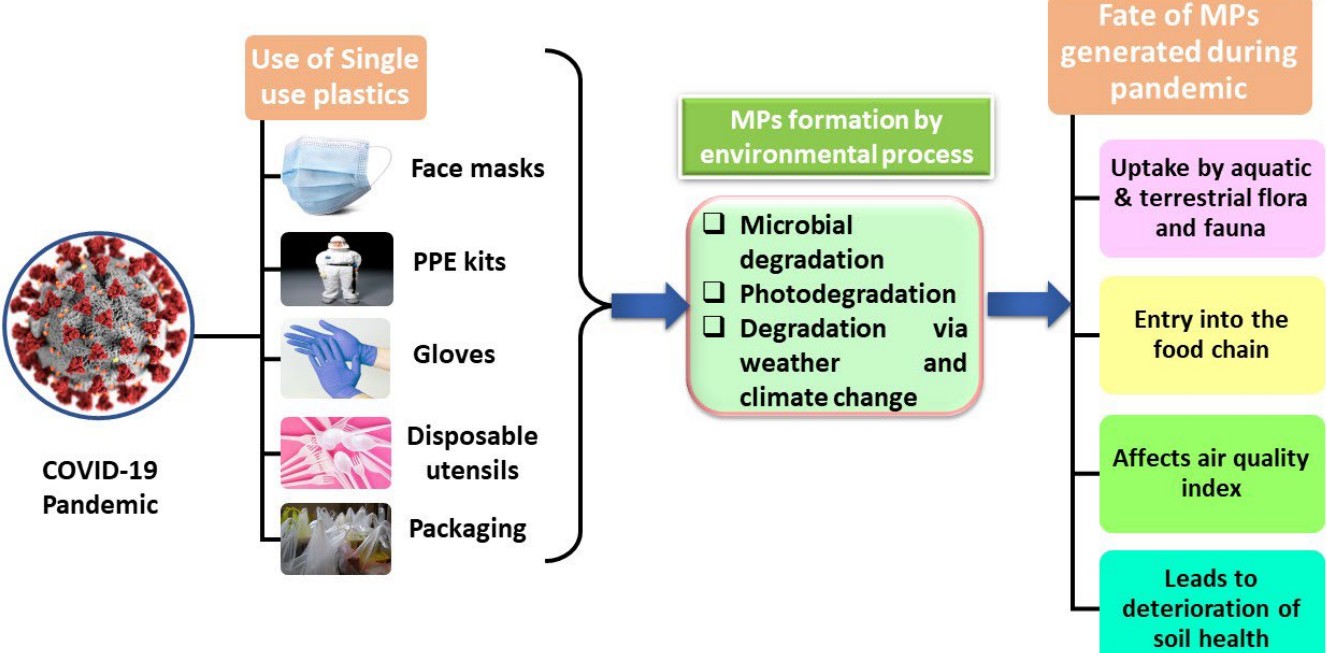

**Figure 3.** Sources, environmental processing, and fate of MPs generated during COVID-19.

Consequently, overloading the existing facilities might lead to paralysis of the waste disposal and recycling industry as a result of this sudden increase in waste. The mismanagement in the disposal of plastic waste might cause MPs to accumulate in terrestrial and marine ecosystems [39]. Hence, pandemics such as COVID-19 pose a serious threat to humankind, and in order to combat their outbreaks the use of PPE kits, face masks, and other polymer products cannot be avoided, leading to the discharge of excessive plastic waste into the ecosystem. Aquatic organisms and terrestrial plants easily accumulate the released MPs from the water and soil, allowing them to be readily consumed by humans and ultimately enter the food chain [40]. Therefore, in order to reduce MP pollution caused by improper disposal of face masks and PPE kits, environmental awareness about proper waste disposal should be implemented as a part of long-term preventive measures.

This study explains how microplastics are generated, transported, disposed and quantify in the environment based on their sources and physicochemical properties. Moreover, the quantification techniques and methodologies for microplastic removal from aquatic systems have been briefly discussed. Additionally, the study also examines the impact of COVID-19 on global plastic waste production. This review aims to gain a better understanding of research on the toxic effects of microplastics on humans, aquatic life forms, and soil ecosystems.

## 2. Detection and Identification of MPs

Due to the exorbitant usage of commodity plastics worldwide, MPs with a wide array of attributes are produced and the detection and analysis of MPs is a prerequisite condition for their effective removal from aquatic systems. A wide variety of physical and chemical techniques, presented in Figure 4, are employed for the quantification of MPs since reliability on a single identification method poses a risk of skipping or missing out on some categories of MPs. Physical detection is frequently used as a preliminary step for easy, low cost and rapid detection of MPs based on their appearance, color, and size. Physical detection does not effectively remove small-sized MPs but is useful for the identification of colored and larger MPs (>500 μm). Therefore, chemical approaches are used to identify the composition and structure of MPs. These include destructive and non-destructive techniques such as SEM-EDX, PLM, FTIR, Raman spectroscopy, and GC-MS.

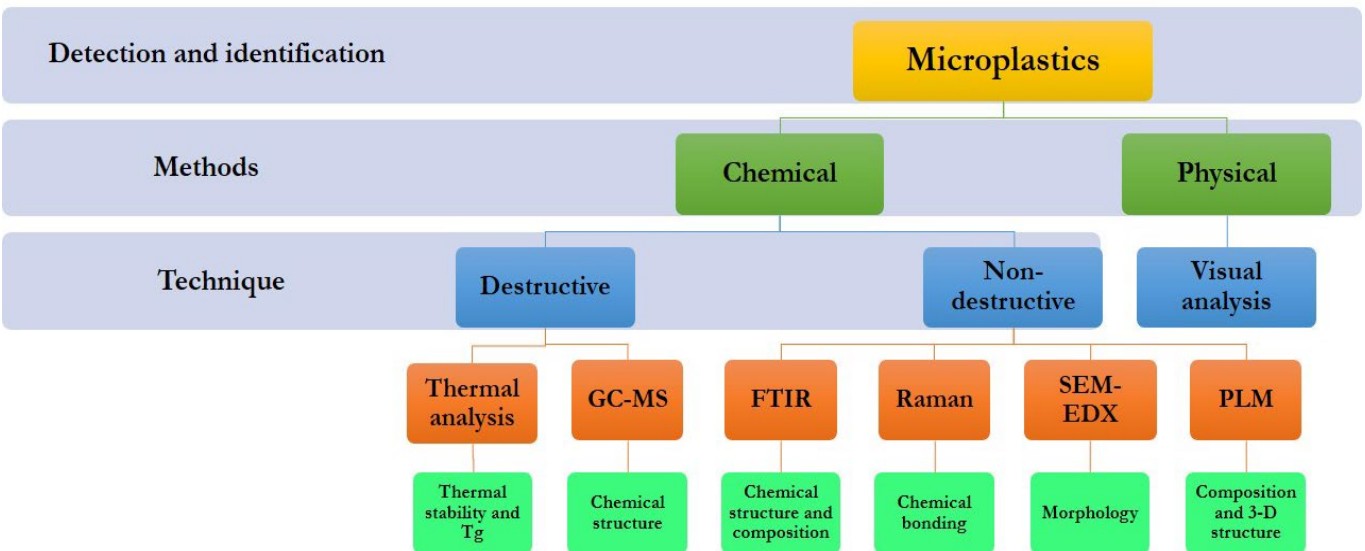

**Figure 4.** Methods for detection and identification of microplastics in aquatic systems.

### 2.1. Identification of Morphology

Scanning Electron Microscopy (SEM) is often used for the morphological analysis of MPs since it captures high-resolution images of MP surface, thereby providing information about the surface texture and deformities that helps in distinguishing MPs from other materials present in the wastewater. This technique is often used in conjunction with the Energy Dispersive X-ray spectroscopy (EDX) technique for the analysis of constituent elements of MPs. There are certain limitations associated with SEM-EDX, i.e., high cost, low efficiency, and inability to detect colored MPs that hinder its applicability for MP detection. To improve the MP detection ability of SEM-EDX, MPs are often stained with fluorescent dyes such as Safranin T, Nile Red, and fluorescein isophosphate at high temperatures to reduce error probability.

The development of advanced microscopic techniques such as Polarized Light Microscopy (PLM) has proved to be quite efficient and useful in the determination of the type of MP. The PLM technique takes advantage of the anisotropic property of polymers and involves the passage of unpolarised light through MP particles that are placed between cross-polarizers. The polarized light emitted from the polarisers imparts information about the crystallinity of MPs and hence aids in the identification of MP polymer type. It is a reliable technique but cannot be used with thick and opaque samples.

### 2.2. Identification of Chemical Structure and Composition

To further improve the identification of MPs and to determine the chemical composition of MPs, certain destructive and non-destructive techniques are employed. FTIR is used

for the detection of IR-active MPs by irradiation of the samples with infrared radiation and noting the changes in the dipole moments of the structural chemical bonds present in the sample. A comparison of the obtained sample spectrum with reference spectrums provides information about the composition of the MPs. Although FTIR is a suitable method for the identification of agglomerates and smaller particles, its functionality for the detection of larger particles is hindered due to limitations associated with sample size, difficult sample preparation, and the labor and time-intensive nature of this technique. To improve the detection efficiency, the FPA-FTIR (Focal Plane Array-Fourier Transform Infrared spectroscopy) technique is employed since it provides a larger spectrum for the MP particles.

The drawbacks of FTIR can be overcome by using Raman spectroscopy based on the principle of inelastic light scattering by polarized molecules. It provides images of MP particles with finer spatial resolutions of 1 um, better than that of FTIR and the results remain unaffected by the thickness and shape of the MPs. It can be used for the identification of non-polar functional moieties and the detection efficiency can be improved by the addition of fluorescent tools as it is a highly sensitive technique. Contamination of samples with dyes, inorganic, organic, and microbial materials strongly impacts the results of Raman spectroscopy. The usage of evolved techniques such as surface-enhanced Raman spectroscopy and Raman tweezers can further improve the detection accuracy for MPs.

### 2.3. Identification of Thermal Properties and Chemical Bonding

Destructive identification techniques such as TGA (Thermo gravimetric Analysis), DSC (Differential Scanning Calorimetry), and GC-MS (Gas Chromatography-Mass Spectrometry) can be used as alternatives to spectroscopic methods for MP identification. TGA and DSC are used for the determination of the polymers on basis of their thermal stability and the glass transition temperature, which varies for each polymer type. The TGA and DSC plots obtained on the thermal treatment of samples are compared with the reference plots to identify the MP type and its characteristics. GC-MS is another popular and reliable technique used for polymer identification in bulk mixture samples. This technique can even be used for nanosized plastics with ease and the detection accuracy can be significantly increased by treatment of samples at elevated temperatures. This is conducted in the TD-GC-MS and pyro-GC-MS techniques. They involve the high-temperature degradation of bulk samples, followed by their segregation via gas chromatography and the subsequent analysis using mass spectrometry. These techniques offer high precision, and sensitivity and can provide qualitative and quantitative results. However, the reproducibility of results highly depends on the sample purity, sample preparation, and thermal treatment conditions. Hence, even with the myriad of methods available for the quantification of MPs, each comes with its drawbacks. Thus, there is still scope for the optimisation of detection and identification methods of MP particles. Further, the feasibility of the usage of chemical methods for MP detection still needs in-depth investigation since the interaction and accumulation of MPs on other materials may strongly influence the detection and identification capability of the above-mentioned methods.

### 3. Removal Methods

The rising level of microplastics in our surroundings poses an imminent threat to human and ecosystem health. Hence, there is an urgent need to devise methodologies for the detection and removal of MPs in order to prevent their bioaccumulation. A variety of physicochemical and biological methods have been devised for MP removal and these methods can be classified into three categories (Figure 5):

(i)     Filtration and segregation
(ii)    Surface adhesion and growth
(iii)   Deterioration

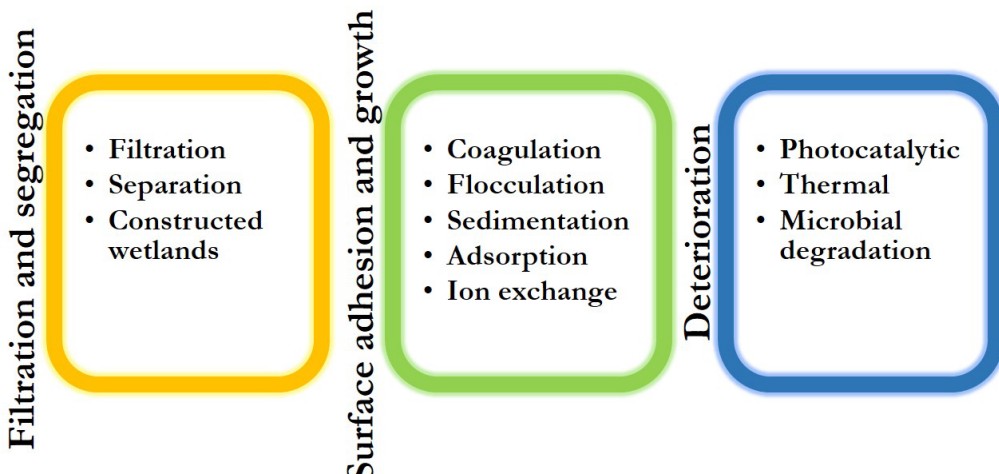

**Figure 5.** Classification of methods used for the removal of microplastics from aqueous media [41–46].

- Filtration and segregation methods: These methods involve the separation of MPs from contaminated water by physical barriers such as membranes and filter mechanisms. These physical barriers only allow the passage of liquids, thereby separating microplastics from aqueous media. However, these methods are often found to be ineffective in the removal of microplastics from sludge waste with higher viscosities. In addition, filtration methods require intensive manpower and require the movement of enormous quantities of water for the separation of micro- and nano-sized microplastics present in minimal concentrations. By using these methods, we only obtain information about the quantification of separated microplastics and do not gather any information about microplastic pollutant type and structure. To obtain detailed information about the type and structure of MPs, we need to adopt other characterisation techniques [47].

- Surface adhesion and growth methods: This method involves the capture and attachment of MPs onto the surface of the added materials (e.g., coagulants, disinfectants, oxidants, surfactants, etc.), causing them to form macrostructures such as aggregates, facilitating their easy removal. This methodology utilizes techniques such as coagulation, flocculation and sedimentation (CFS), adsorption, and ion exchange. Unlike the filtration and segregation methods, these methods are efficient, easy to handle and monitor, and are even helpful in the removal of other pollutants. However, due to a lack of information, they are still only performed at the pilot scale instead of large-scale operations. However, these methods possess certain limitations, i.e., they are often time-intensive and ineffective for the uptake of smooth, small-sized microplastics due to a lack of sufficient surface area to either adhere to the surface of the added materials or form flocs [48].

- Deterioration methods: Another method used for the separation of microplastics is the deterioration method which makes use of the action of external factors such as radiation, heat, and microorganisms to bring about changes in the physiological structure of MPs and break them down into simpler molecules such as $CO_2$, $H_2O$, $H_2S$, methane, etc. Photocatalytic, thermal and microbial degradation fall under this category. Degradation methods are one of the most efficient methods for combating MP waste but these methods are not much explored and still need further in-depth studies for understanding the detailed mechanisms involved in degradation to fully exploit their potential. The breakdown capacities efficiencies can also be enhanced which can ultimately lead to a reduced degradation time span [49]. Table 2 presents the advantages and disadvantages of the above-mentioned removal methods.

**Table 2.** Advantages and disadvantages of methods employed for microplastic removal from aqueous media.

| Removal Method | Advantages | Disadvantages |
|---|---|---|
| Filtration [41] | • High removal efficiency<br>• Stable effluent quality<br>• Easy to handle | • Membrane fouling<br>• Possibility of secondary MP formation<br>• Frequent cleaning required |
| Constructed wetlands [42] | • Less maintenance<br>• Low operating cost | • Little information about the mechanisms involved<br>• Influence of external factors not fully understood |
| Coagulation-Flocculation-Sedimentation [43] | • Simple and easy to operate<br>• Ability to capture and remove small-sized microplastics | • High requirement for chemicals<br>• Majorly studied only in laboratories<br>• Not widely studied at the commercial level |
| Adsorption and ion exchange [44] | • Recyclability of adsorbents and ion exchangers<br>• Can remove MPs less than 100 µm | • Large time spans for adsorption and ion exchange required<br>• Handling adsorbents may be difficult |
| Photocatalytic degradation [45] | • No requirement for chemicals<br>• Environment-friendly<br>• Applicable for multiple MPs types such as PE, PS, PET, etc. | • Low efficiency<br>• May produce harmful by-products<br>• No selectivity |
| Microbial degradation [46] | • Low cost<br>• Simple and flexible usage | • Less information available<br>• High degradation time<br>• May produce secondary MPs |

Since the above-mentioned methods are often unable to efficiently and completely remove microplastics when used singularly, they are used in conjunction with each other, thereby forming the primary, secondary, and tertiary stages of wastewater purification (Figure 6). Some of these removal methods are described below in the following sub-sections.

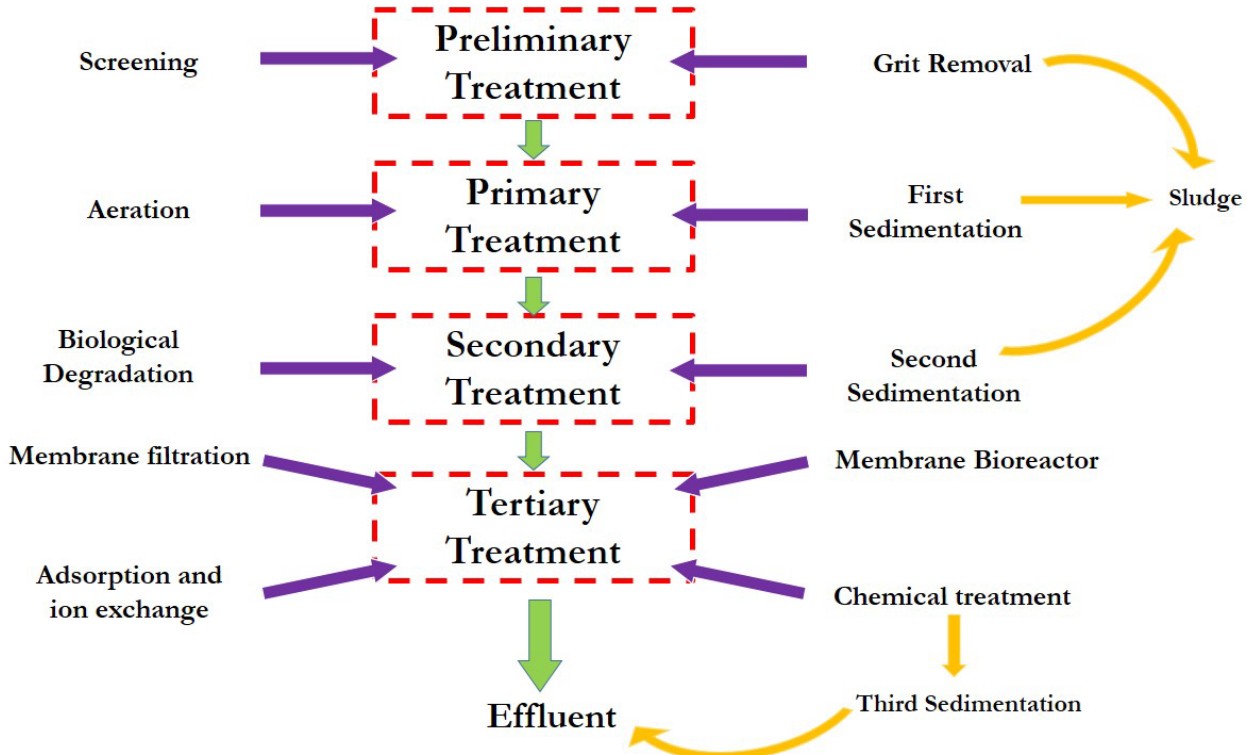

**Figure 6.** Schematic representation of methods and stages in wastewater treatment.

### 3.1. Membrane Filtration

Membrane filtration is an advanced technique that has been recently developed especially for MP removal and involves the movement of polluted water across a membrane with pore sizes varying according to variation in the shape and size of MP particles. The use of membranes is being adopted increasingly due to the low energy requirements, facile and flexible operation, easy scalability, and stability of the method. Membrane filtration is an umbrella term and is used for multiple methods such as ultrafiltration, microfiltration, nanofiltration, and, reverse osmosis [50–53]. Membrane bioreactors (MBR) and dynamic membrane (DM) systems have also been established for efficient MP removal. Membranes with varied pore sizes and external conditions such as pressure, and pumping shear stress help in the effective operation of these systems. The permeability and selectivity of membranes, their durability, the size and concentration of MPs, and influent flux are key factors that influence MP removal efficiency by a large degree. Membrane processes have been known to exhibit removal efficiencies of up to 99.9% when used in combination with other techniques. For instance, Lares et al. (2018) devised an advanced MBR system incorporated with WWTP to analyze MP removal. They compared the MP removal efficiency of the MBR device with a conventional activated sludge method and concluded that MBR showed higher removal efficiency towards MP removal than the conventional activated sludge method [35]. Tadsuwan et al. [54] investigated the outcome of coupling ultrafiltration with a pre-existing water treatment plant and reported an increase in removal efficiency from 86.14% in a traditional water treatment plant to 96.97% combined with an ultrafiltration setup. Li et al. [55] formulated a dynamic membrane (DM) on a 90 µm mesh via synthetic wastewater filtration and investigated the impact of control parameters on the functioning of the DM. They concluded that the DM was formed rapidly and was quite effective in MP removal, and it was promoted by the increased motion of solids and concentration of influent particles [55].

The membranes used in membrane filtration often suffer from fouling phenomenon caused by the interaction and accumulation of MPs in the membrane pores and the growth of microorganisms on them, leading to their clogging, and thereby reducing their removal capability (Figure 7). Therefore, a need often arises to pre-treat the polluted water with disinfecting agents or coagulants to prevent this phenomenon; although pre-treatment of water often reduces the probability of membrane fouling. A study by Xing et al. describes a low-dosage UV/Chlorine pre-oxidation strategy to prevent membrane fouling. They reported the pre-oxidation method was for reducing membrane fouling by 49% [56]. It is also suspected that cleaning and backwashing of membranes may aggravate the problem of MP release. Membrane filtration is quite successful in the removal of fragment and pellet MPs due to synergistic interactions between the MP particle and membrane material and pores. However, they are less useful for fiber MPs since fiber-type MPs may move longitudinally through the pores of the membrane. Ziajahromi et al. [57] prepared an MP sampling device and studied the microplastics present in effluents released from three different WWTPs that used treatment methods such as CFS method, biological treatment, disinfection/de-chlorination processes, and ultrafiltration, followed by reverse osmosis (RO) process. They detected the presence of microplastic fibers in the effluents even after reverse osmosis and attributed it to the existence of membrane defects [57]. Thus, an in-depth study of the MF methods is required to enhance their efficiency for MP removal at the pilot scale.

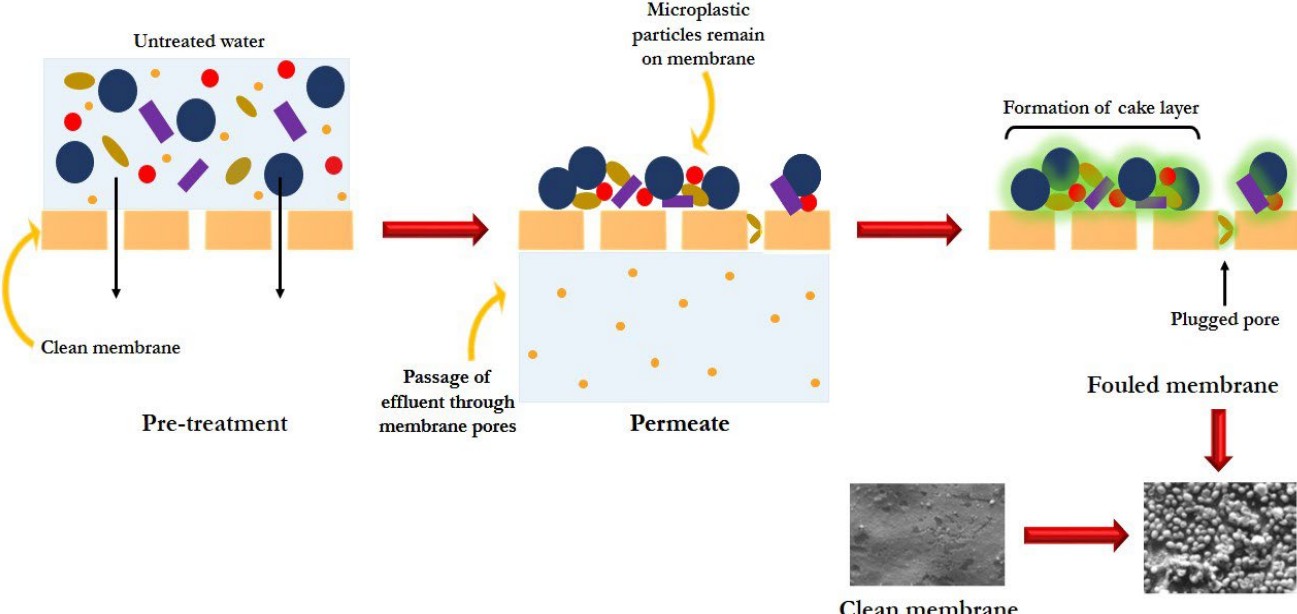

**Figure 7.** Schematic representation of microplastic removal by membrane filtration and membrane fouling [58].

*3.2. Adsorption*

The adsorption method takes precedence over many other methods for pollutant removal from aqueous media due to its facile nature, high efficiency, economical usage and other advantages. Adsorption is a surface phenomenon and involves the uptake of pollutants on the adsorbent surface by means of weak Van Der Waal interactions. A variety of materials such as carbonaceous materials, zeolites, polymers, and inorganic clays have been utilized for the removal of MPs from water sources. The high number of adsorption sites, nature, and strength of the adsorbent are deciding factors in the adsorptive and regenerative capabilities of adsorbent materials. Tang et al. synthesized magnetic carbon nanotubes and reported the adsorption of PE, PET, and PA microplastic particles. The adsorbent exhibited 100% removal efficiency for the MP particles and showed maximum adsorption capacity for PE, then PET and least for PA, and exhibited <80% removal efficiency even after four adsorption cycles [59]. Recently biosorption has emerged as a viable and effective method for MP uptake (Figure 8). The use of biomass, bacteria, fungi, algae, seaweed and other industrial and agricultural biowaste is being highly favoured as it does not lead to the discharge of secondary MPs into water bodies. The adsorption process with such biomaterials generally proceeds via physical adsorption, ion exchange, chelation, microprecipitation and complexation mechanisms in the extracellular technique. The presence of hydroxyl, amine, carboxyl, and phosphonate groups in the cell walls of microbes and plant bodies aids in microplastic adsorption. In a recent study by Sundbæk et al. [36], the marine algae Fucus vesiculosus was used for the adsorption of MPs. The constituent carboxylic groups of alginic acid present in algae cell walls are responsible for the binding of MPs to the adsorbent surface. A detailed report by Siipola et al. showed the usage of steam-activated pine and spruce bark-based biochar for the purification of urban wastewater and runoff. They focused on determining the effect of features of biochar adsorbent, such as chemical composition and particle size, and concluded that these bioadsorbents were suitable even for the removal of very small-sized microplastic particles. They performed the mechanism for retention of MP particles on bioadsorbent surfaces and still more research needed to be conducted to gain a deep understanding of the adsorption mechanism [60]. Sun et al. fabricated chitin and graphene oxide-based compressive sponges that can adsorb MP particles at pH 6–8 with a high removal efficiency of 89.7% and offers recyclability for up to three cycles. The sponge was also found to be

biocompatible as it did not inhibit the growth of green algae on its surface and could be broken down by microorganisms present in the soil, confirming its biodegradability [61]. Although adsorption is an easy and effective method, drawbacks such as time and labor intensiveness limit its advantageous usage. Thus, it may be used in combination with other advanced techniques for better MP removal efficiencies.

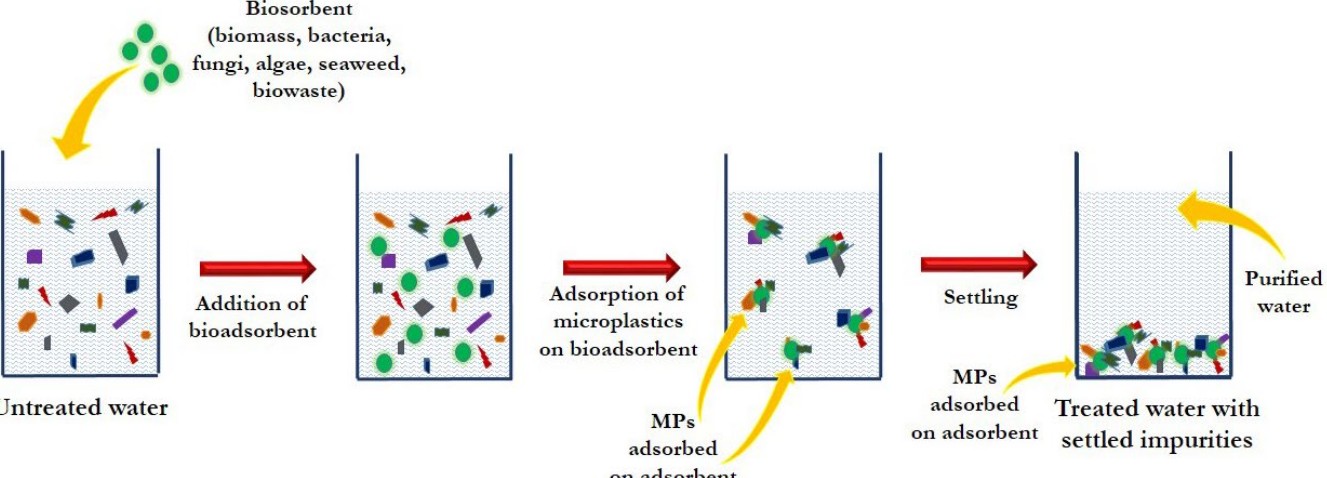

**Figure 8.** Schematic diagram illustrating the bio-adsorption of microplastics.

### 3.3. Coagulation, Flocculation and Sedimentation (CFS)

The combination of coagulation, flocculation and sedimentation is the most widely used method for MP uptake from water sources. The CFS mechanism involves the heterogeneous separation of solids and liquids and is monitored by the density of solids and liquids (Figure 9). Coagulation involves the destabilisation of suspended particles in a colloid by the addition of a coagulant material such as metal salts. It is a rapid method often followed by flocculation. The flocculation technique involves slow mixing for long time intervals, leading to the aggregation of previously destabilized particles to form large aggregates (flocs) that can then be easily removed by sedimentation. Flocs formed during stirring are influenced by the characteristics of aqueous media such as ionic strength, pH, divalent cations, natural organic matter, and particulate/colloidal matter. Zhang et al. [62] synthesized a magnesium hydroxide-$Fe_3O_4$-based magnetic coagulant and applied it for microplastic removal. They reported a removal efficiency of 87.2% and also explored the influence of aging time on floc formation. They also concluded that removal efficiency above 85% can be maintained in water samples within the pH range of 5–8 and that charge neutralisation is an important mechanism involved in microplastic removal [62]. The sedimentation technique is based on the gravitational settling of suspended aggregates and is impacted by microplastic particle density. It is especially helpful in the elimination of irregularly shaped MP fragments since angular and irregularly shaped particles can be easily captured to form bigger aggregates that can settle down due to increased density. These methods are often used together to enhance MP removal capacities and the removal efficiency of the CFS method is monitored by the physiochemical and morphological properties of MPs, i.e., shape, size, and surface properties. These methods are often employed as primary or secondary treatment methods and are often used in conjunction with other advanced techniques to maximize MP removal efficiency. The CFS method is more successful in removing fibrous MPs than spherical or fragmented MPs due to the availability of a larger surface area, facilitating more interaction with flocculating agents. Peydayesh et al. (2021) investigated the uptake of carboxylated PS microspheres from various water samples using a lysozyme amyloid fibril natural bioflocculant by CFS technique with 98.2% removal efficiency [37]. Pivokonský et al. tested raw and purified water from two different drinking water treatment plants (DWTPs) and identified that the CFS method is quite effective in

eliminating microplastic particles from the water samples [63]. Lapointe et al. examined the performance of the CFS method for the sequestration of pristine and weathered PE, PS and PEST microplastic particles and noted that the removal efficiency was found to be maximum at 97% for weathered PEST particles. They also explored the use of settled water turbidity as a possible indicator for the removal of MPs [43]. The CFS method suffers from drawbacks such as high chemical consumption, large power requirements, and frequent electrode replacement, limiting their cost-effective usage for water purification.

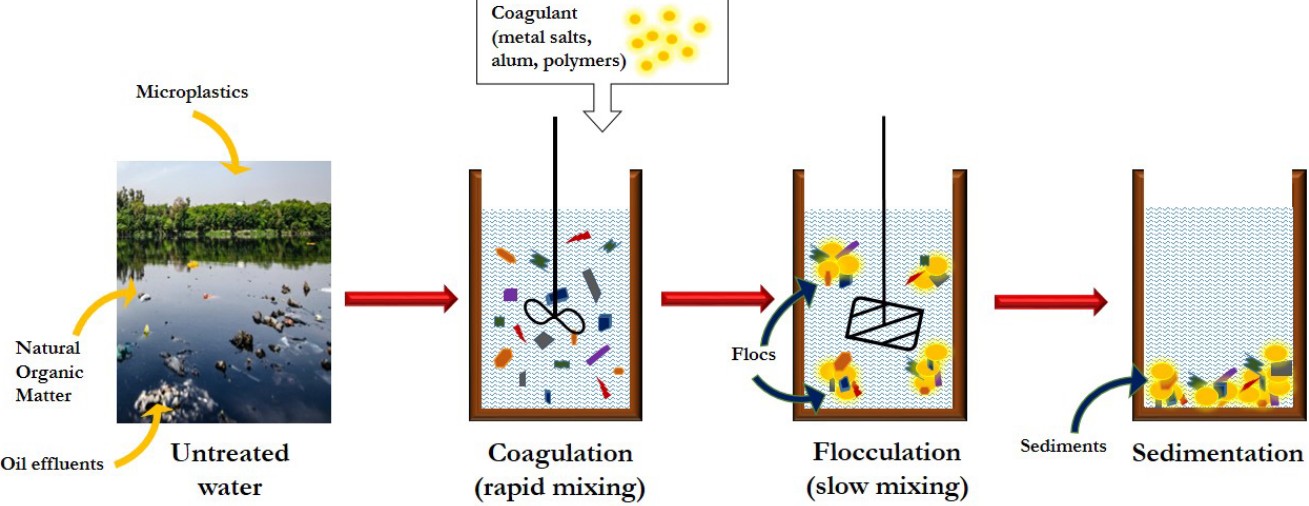

**Figure 9.** Microplastic removal by coagulation, flocculation and sedimentation processes.

*3.4. Biological Degradation*

The biodegradation of MPs is an environmentally benign method for the eradication of MPs from aqueous systems. A variety of microbes such as fungi, diatoms, bacteria, biofilms, etc., are reported to induce the degradation of PE, PP, and PS microplastics. These microbes grow and form colonies on the MP surface and use them as carbon sources, thereby leading to their degradation, producing secondary MPs, $CO_2$, methane, $H_2O$, and biomass (Figure 10) [64]. The biodeterioration process may be aerobic or anaerobic, depending on the presence of oxygen, and is influenced by external factors such as temperature, humidity, UV and solar radiation. The efficacy of biodegradation mainly depends on the type of polymer, its characteristics and morphology, and the molecular weight. Auta et al. [38] studied the impact of isolated *Bacillus gottheili* bacteria on the deterioration of PE, PET, PP, and PS over a span of 40 days [38]. They observed changes in the surface texture as well as the formation of grooves and cracks and concluded that bacteria affect the surface and bulk properties of MPs. The marine fungus *Zalerionmaritimum* was used by Paço et al. [39] for the degradation of PE pellets and they observed molecular changes in the MPs along with a reduction in PE pellet mass and size, confirming their degradation. Biodegradation of PVC microplastics using the larvae of *Tenebrio molitor* was investigated by Peng et al. and they observed partial biodegradation of polymer along with the formation of smaller chlorinated organic compounds and reduction in $M_n$ by 32.8%. They concluded that the ingestion rate was slow and the mineralisation of PVC microplastic powder was only partial [65]. Huang et al. [66] surveyed the distribution of microplastics on biofilms consisting of filamentous algae in the Middle Route of the South-to-North Water Diversion Project (SNWDP) in China, a regulated canal and reported that MPs were concentrated on the biofilms, with small PET fibers being the major category MPs present in the biofilms. They concluded that these biofilms could be used as a sink for microplastics. Although, the usage of microorganisms for MP degradation is a flexible and tunable technique, the time duration required for these methods is substantial. Additionally, there are challenges in scaling up these methods, and they are suspected to release secondary MPs along with

the non-reusability of the microbes. The degradation mechanisms involved require further in-depth investigation to fully utilize the potential of microbes for MP degradation.

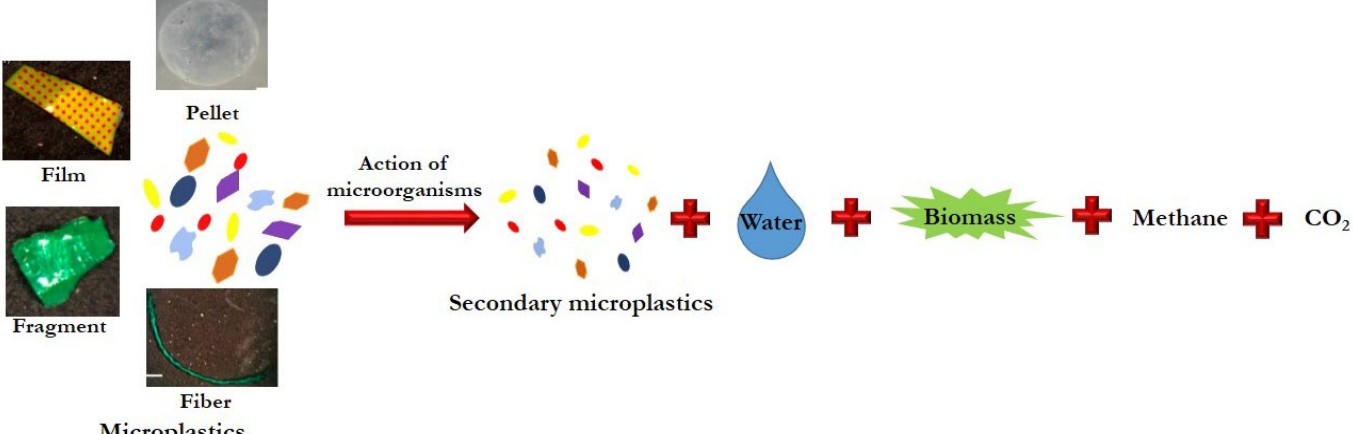

**Figure 10.** Microbial degradation of microplastics [67].

*3.5. MP Shape, Size, and Polymer Type and Their Impact on Efficiency of Removal Methods*

The shape, size and polymer type of MPs have a vital impact on the effective extraction since particles with different shapes, sizes and compositions exhibit varying properties and toxicities. Shape and size are helpful in the analysis of accumulation patterns of MPs in water bodies since low-density MP particles, such as fibers and fragments, tend to stay afloat on the water surface while MP particles such as pellets that possess relatively higher densities sink to the bottom of water bodies. Due to their ubiquitous presence, there arises the need to understand and identify the transportation and dispersal mechanisms across soil and water systems.

### 3.5.1. Impact of Shape of MP

MPs occur in many shapes in aquatic systems but the majority of MPs exist as fragments, pellets, spheres, films, and fibers. Among these forms, fragments and fibers are the most prevalent in water bodies [40,41]. The biggest contributor of fiber MPs to water bodies is the garment industry, leaching fiber-type MPs into aquatic systems via effluents released from their washing processes. Film-type MPs usually originate from the weathering of packaging materials and plastic bags while the usage of MP pellets for abrasion applications in the cosmetics industry contributes to their release. Fragment MPs are usually secondary MPs and often arise from the degradation of bigger plastic objects. It has been observed that fiber MPs are typically eliminated in the primary treatment step by techniques such as coagulation, flocculation and sedimentation (CFS). While during the secondary treatment stage, maximum removal of fragment-type MPs occurs since their lamellar structure facilitates their agglomeration and subsequent removal. Substantial removal of MP pellets is seen in the tertiary treatment stage, involving the use of advanced filtration and oxidation techniques. The tertiary treatment processes are most favoured for the removal of MPs with very small sizes and have distinct features. However, it was noted that the MP removal efficacy of tertiary treatment stages is lesser than those of primary and secondary treatment stages.

### 3.5.2. Impact of Size of MP

The size of MP particles also influences the determination of removal efficiency. During the primary treatment processes, large MP particles having sizes larger than 0.5 mm are easily removed by methods such as flocculation, sedimentation and grease removal. Large-sized fiber and film-type MPs are easily removed by flocculation and grease removal methods due to their low densities while small MP pellets sink to the bottom of containers

due to action of gravity on these high-density particles. Additionally, due to weathering phenomena, primary microplastics often break down into secondary MPs that are ingested by aquatic organisms and lead to bioaccumulation and toxicity. Thus, the removal of secondary MPs holds utmost importance but they cannot be extracted by conventional methods, thereby requiring the use of sophisticated techniques and instrumentation for their efficient removal.

### 3.5.3. Impact of Polymer Type of MP

The ubiquitous utilisation of PE, PP, PET, PU, PA, PAAm, PVC, PES, PS, PEVA and other such polymers can be ascribed to their versatile properties and stability. Among these, PE and PS are the most highly favoured materials due to their excellent impact and chemical resistance, low production costs, and easy workability, thus finding application in multiple industries. Due to their vast usage, the proportion of PE and PS MPs present in aquatic systems is much more than other commodity MPs. PE and PS MPs are positively charged, and hence they can be effectively removed from wastewater by using secondary treatment processes. Therefore, the impact of the morphological attributes and types of polymers on the efficient removal of MPs needs to be studied in detail. It will also help in gaining a deeper understanding of the removal mechanisms by conventional and advanced MP removal technologies.

### 4. Accumulation of MPs in the Ecosystem and Their Toxicity Assessment

The environment can be affected by MPs in a variety of physical, chemical, and biological ways. Various physical injuries may occur to animals when they become entangled in microplastics in the environment, including drowning, suffocation, strangulation, and starvation [68]. The chemical impact of MP on the environment is attributed to the adsorbed chemicals onto plastic surfaces. MPs are composed of highly hydrophobic materials, making them a potentially toxic chemical reservoir. Furthermore, the presence of excessive levels of MPs in ecosystems can also impair the normal physiological functioning of living organisms [27]. Throughout the food web, MPs may pose an environmental risk because of their bioavailability. Aquatic and terrestrial environments contain a high concentration of MPs, which would be present in food products consumed by humans.

### 4.1. Impacts of Microplastics on Human Health

A recent report released on Microplastics in Drinking-water (2019) by World Health Organisation highlighted the ubiquitous presence of MPs in the ecosystem and raised concerns about their adverse effects on human health [69]. Growing concern over microplastics' potential health impacts has been raised since microplastics can enter the human food supply via the ingestion of terrestrial foods and seafood. The existence of MPs in foods consumed by humans has also been highlighted by many groups (Table 3).

**Table 3.** Presence of MPs in food items and drinks consumed by humans.

| Consumable Products | Polymer Types | Size | MPs Concentration | References |
|---|---|---|---|---|
| *Seafood* | | | | |
| Bivalve (oyster, mussel, Manila clam, and scallop) | PE, PP, PS, PES, PEVA, PET, PUR | 0.1–0.2 mm | 0.97 (0–2.8) particles/individual 0.15 (0–1.8) particles/g | [70] |
| Canned Sardines | PE, PET, PVC, PP | 190–3800 μm | 6 MPs per item | [71] |
| Fish | PET, PP, PUR, PES | <500 μm | 2.2 ± 0.89 MPs/individual | [72] |
| *Acanthopagrus australis* (Yellowfin bream) | PET, RY, PES | - | Mean 0.6 MPs/fish | [73] |
| Pelagic and demersal fish | Cellulose, PA, RY | 0.13–14.3 mm | 1.90 particles/individual | [74] |
| *Engraulis japonicus* (Japanese anchovy) | PE, PP, PS | 150–1000 μm | Mean 2.3 MPs/individual | [75] |
| *Fenneropenaeus indicus* (Indian white shrimp) | PA, PES, PE, PP | 0.157–2.785 mm | 0.39 ± 0.6 items/shrimp 0.04 ± 0.07 items/g | [76] |
| *Mytilus edulis* (Mussels) | CPH, PET, PES PE, | 0.033–4.7 mm | 0.9–4.6 particles/individual 1.5–7.6 particles/g | [77] |

**Table 3.** *Cont.*

| Consumable Products | Polymer Types | Size | MPs Concentration | References |
|---|---|---|---|---|
| *Meat* | | | | |
| Poultry, cows, and pigs | PP, PE, PET | <5 mm | Poultry manure: 667 ± 990 particles/kg<br>Cow manure: 74 ± 129 particles/kg<br>Pig manure: 902 ± 1290 particles/kg | [78] |
| Chicken gizzards | PS | 0.1–5 mm | 10.2 ± 13.8 particles/g | [79] |
| *Salts* | | | | |
| Salt | CPH, PE, PET | <200 μm | Lake salt: 43–364 particles/kg<br>Rock salt: 7–204 particles/kg<br>Sea salts: 550–681 particles/kg | [80] |
| Sea/lake/rock salt | PE, PET, PP | <500 μm | Lake salt: 28–462 particles/kg<br>Rock salt: 0–148 particles/kg<br>Sea salts: 0–1674 particles/kg | [81] |
| *Drinks* | | | | |
| Tea | PA, PET | 25 μm | ~11.6 microplastics/cup of the beverage | [82] |
| Drinking water | PET, PE, PA, PP | 0.005–0.1 mm | 1 ± 8 particles/L (beverage cartons)<br>118 ± 88 particles/L (returnable plastic bottles) | [83] |
| Milk | Polysulfone | 0.1–5 mm | 6500 particles/m$^3$ | [84] |
| Drinking water | PES, PVC, PE, PA, EP | 0.05–0.105 mm | 0–7000 particles/L | [85] |
| Beer | - | 0.1–5 mm | 0–14.3 particles/L | [86] |
| *Sugar and honey* | | | | |
| Honey | PP, PE, PAAm | 0.013–0.25 mm | 54 particles/L (industrial honey)<br>67 particles/L (craft honey) | [87] |
| Honey | - | 0.01–9 mm | 166 ± 147 particles/kg (fibers)<br>9 ± 9 particles/kg (fragments) | [88] |
| Sugar | | | 217 ± 123 particles kg$^{-1}$ (fibres)<br>32 ± 7 particles kg$^{-1}$ (fragments) | |

Note(s): PE—Polyethylene, PUR—polyurethane, PP—Polypropylene, PA—Polyamide, PET—Polyethylene terephthalate, PAAm—Polyacrylamide, PES—Polyester, PVC—Polyvinyl chloride, PS—Polystyrene, PEVA—Poly (ethylene-co-vinyl acetate), RY—Rayon, EP—Epoxy resin, CPH—Cellophane.

In addition, MPs are also absorbed into the body when they are inhaled and come in contact with skin [89,90]. Microplastics are ingested mainly through food products such as table salt, mussels, sugar, commercial fish, and even water, which are contaminated with microplastics. MPs may enter the digestive tract, triggering inflammatory responses, increasing intestinal permeability, and altering the metabolism [91]. The toxic effects of MPs exposure in humans are presented in Figure 11.

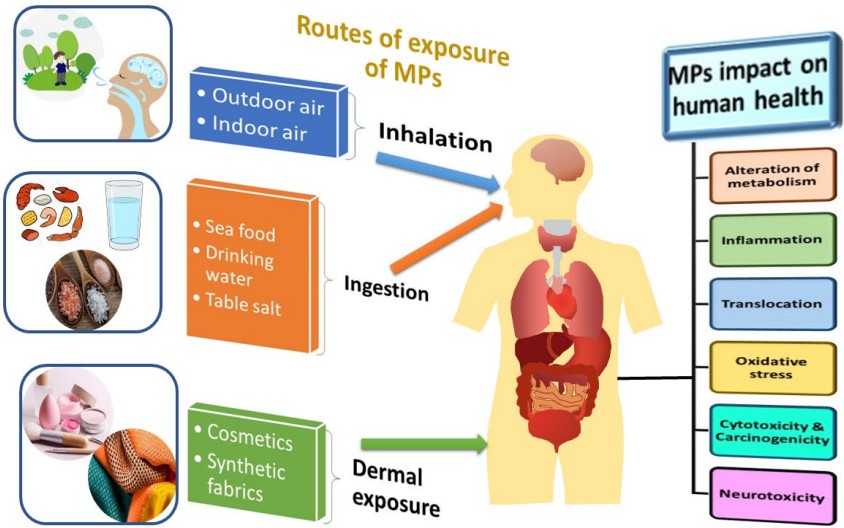

**Figure 11.** Schematic illustration of exposure of MPs on human health.

Various sources release MPs into the atmosphere, including textiles, abrasion of car tires, buildings, etc., and microplastics resuspension from surfaces. Prata (2018) reported that the amount of MPs inhaled by individuals per day ranges from 26 to 130 MPs. Consequently, this could be hazardous to human health because of their polymeric structure which makes their removal from the respiratory system difficult and they release toxic organic pollutants and plasticizers from their surface [92]. The risk of inhalation of MPs from wearing different masks during the COVID-19 pandemic was investigated by Li et al. [58]. They reported that wearing N-95 masks posed a lower risk of inhalation of fiber-like MPs as compared to the activated carbon masks [93].

Currently, almost negligible studies have been conducted to evaluate the associated risks of dermal exposure to MPs in humans. Although the extensive usage of microbeads in personal care products and synthetic fabrics along with the presence of microplastics in indoor dust particles leads to considerable human exposure to MPs via dermal contact. Microplastic beads having a size of less than 1 mm have been extensively utilized in facial scrubs, toothpaste, and dentures [94]. Human exposure to MPs via dermal contact has not been comprehensively studied, a few studies have only assessed the per capita consumption of MPs. A study by Napper et al. [26] showed that usage of facial scrubs by the UK population alone is discharging 40.5–215 mg of polyethylene microbeads person$^{-1}$ day$^{-1}$. Although human skin is susceptible to penetration by particles less than 100 nm in size, microplastics may penetrate through hair follicles, open wounds, or sweat glands to cause skin damage. It is imperative that in-depth research be carried out on human dermal exposure to MPs via cosmetics, settled dust particles, and fabric fibers so that the significance and health risks associated with these exposure routes can be determined [95].

The risk of ingesting microplastics has not yet been quantified completely due to the relatively limited amount of research. Although several groups have performed in vitro studies to assess the toxicological effects of microplastics on human health, there is still a lack of availability of data on in vivo studies. In this section, some of the studies which examined the toxicity of MPs on human cell models have been discussed. The cell viability and cytotoxicity of microplastics in terms of oxidative stress were investigated by Schirinzi et al. [96] on cerebral (T98G) and epithelial (HeLa) human cells. The results demonstrated that in both cases, i.e., with exposure to polyethylene (3–16 μm) and polystyrene (10 μm), cell viability was not affected. Wu et al. [97] studied the cytotoxicity of 0.1 and 5 μm polystyrene MPs on human Caco-2 cell lines. The results indicated that both MPs exhibited weak cytotoxicity and displayed negligible changes in membrane integrity, whereas both sizes lead to disruptions in mitochondrial potential with larger MP sizes producing greater disruption than smaller MP sizes. Stock et al. [98] also studied the cytotoxic effects of polystyrene MPs (sizes ranging from 1, 4, and 10 μm) on the human Caco-2 cells and monocyte-like THP-1 cells. The results indicated that MPs of 1 μm size affected the cell viability of Caco-2 cells. Hesler et al. [99] performed in vitro analysis of the toxicological effects of modified polystyrene (0.5 μm) at the human intestinal and placental barrier. The MPs exhibited no genotoxicity and weak embryotoxicity. In vitro analysis performed by Xu et al. [100] confirmed the cytotoxicity induced by PVC particles (2 μm) in human pulmonary cells. A particle's size and density determine the deposition of MPs into the human respiratory system, smaller and less dense particles penetrate the lungs more deeply. The toxicology of these particles needs to be further researched on a multidisciplinary and international scale in order to understand their long-term impact on humans.

*4.2. Impacts of MPs on Aquatic Environments*

As microplastics (MPs) accumulate in the environment, aquatic life is becoming more vulnerable. Secondary microplastics resulting from the fragmentation of automobile tires, packaging materials, paints, synthetic fabrics, etc., are the primary source of contamination of water resources [101]. Poor waste management is also responsible for introducing microplastics into freshwater through runoff from surface and agricultural areas (Figure 12). It has been identified that wastewater and sewage treatment plant effluent discharges

are a chief source of introducing MPs into the freshwater. As MPs float on the water surface, disperse throughout the column, and accumulate in seawater sediments, they can be consumed by diverse aquatic organisms occupying a variety of habitats [1]. The impact of MPs on aquatic environments can be classified as physical and chemical. Aquatic species can be physically impacted by the MPs by entanglement or by ingestion, with the former being the more common. Allsopp et al. [102] revealed that entanglement results in the suffocation, drowning, or starvation of aquatic animals. It has been reported that ingested plastic fragments have caused physical injuries in animals, including ulcerations and rupture of the digestive tract. Aquatic organisms are incapable of differentiating between MPs and natural prey items resulting in accidental uptake of MPs, leading to an influx of microplastics into the aquatic food web [103].

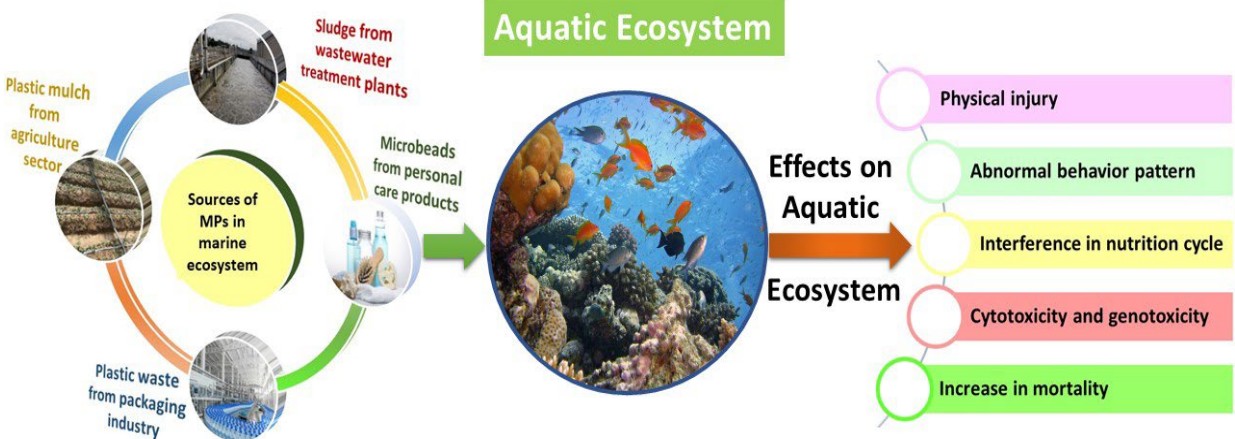

**Figure 12.** An overview of sources and toxic effects of MPs on aquatic environment.

A number of adverse effects may be induced by MPs on aquatic species, such as behavioral changes, slow metabolism, and disruption in growth and reproduction. When aquatic organisms are severely overloaded with MPs, they display lethargic swimming and feeding behavior. Yin et al. [104] showed that the accumulation of MPs in the digestive tract may result in abnormal behavior in fish. A study by Chen et al. [105] revealed that exposure to PS-MPs at 1 mg/L concentration suppressed the catalytic activity of acetylcholinesterase (AchE) on zebrafish larvae. The inhibition of AchE activity subsequently results in the over-stimulation of receptors and may result in paralysis and death as a result of a significant build-up of AchE in synaptic clefts. Consequently, prolonged exposure to MPs could influence the nutritional status of fish, thus affecting their health and growth.

According to Sussarellu et al. [106], polystyrene (PS) MPs adversely affect oyster reproduction and feeding by altering their food intake and energy distribution. The quality of oocytes, motility of sperm, and egg production were all reduced in oysters exposed to microsized polystyrene. The eggs and sperm are released in the sea for external fertilisation by oysters, but because of the intake of MPs, the sperm's speed and count were significantly reduced [106]. Two-month exposures of adult oysters (*Pinctada margaritifera*) to 6 and 10 µm polystyrene microbeads were conducted by Gardon et al. [107] to examine the effect of MP on their physiology. *P. margaritifera*'s assimilation efficiency, energy balance, and reproduction were significantly affected by PS-MPs. Cole et al. [108] examined how MP consumption affects the fertility, feeding habits, and functioning of the copepod *Calanus helgolandicus*, which because of its size, lipid content, and opulence, is a vital prey species for many fish larvae. As a consequence of exposure to 20 µm PS microbeads, the carbon biomass of copepods was reduced by 40%, resulting in an energy reduction and increased consumption of lipids, which affected their growth [108]. Banaee et al. [109] studied the effect of polyethylene MPs on various biochemical parameters of blood on *Emys orbicularisto* (pond turtle). The MPs exposure adversely affected all the parameters studied which indicated liver and kidney dysfunction [109]. As a result of exposure to PS

microplastics, *Danio rerio*'s (zebrafish) metabolic pathways were changed and its lipid and energy metabolism was altered [110]. In addition, Lei et al. studied the toxic effects of five varieties of MPs on Zebrafish and reported that *D. rerio* was observed to develop intestinal damage after exposure to microplastic particles. Moreover, they also demonstrated that MPs' lethality is not dependent on their chemical composition, but on their size.

In a study by Kaposi et al. [111] PE microspheres (10–45 μm) were exposed to *Tripneustes gratilla* (sea urchin) larvae for 5 days. They concluded that although a significant decrease was observed in larval body width, the current MP levels in the ocean present only a limited threat to marine invertebrates. Weber et al. [112] indicated that despite high levels of MP contamination, Gammarus pulex (amphipod) did not show significant effects on development, survival, feeding behavior, or metabolism (glycogen, lipid storage). Zhang et al. [62] assessed the toxic effects of variably sized PVC MP on *Skeletonema costatum* (marine algae). The results suggested that small-sized PVC MPs adversely affected photosynthesis and inhibited the growth of microalgae. PVC MPs have been exposed to marine *Perna Viridis* (Asian green mussels) by Rist et al. [113]. It was observed that there was an increase in mortality after MP exposure, with decreased filtration and respiration rates, as well as reduced motility [113]. A study by Rochman et al. [114] demonstrated the impact of short-term exposure to PE fragments in marine *Oryzias lapites* (Japanese medaka fish) and observed that this led to the bioaccumulation of chemicals and early tumour development. When *Arenicola marina* L. (lugworms) was exposed to a high concentration of PVC MPs, the immune function of lugworms was impaired and a high mortality rate was observed in a study by Browne et al. [115]. Table 4 summarizes the toxic effects of some of the MPs on different aquatic species.

**Table 4.** Effects of MPs on aquatic organisms.

| Aquatic Organisms | Polymer Types | Size | Effects | References |
|---|---|---|---|---|
| Zebrafish Larvae | PS | 45 μm | Suppressed catalytic performance of AchE | [105] |
| Oyster | PS | 2 and 6 μm | Reduced sperm count and speed | [106] |
| *Pinctada margaritifera* (Oyster) | PS | 6 and 10 μm | Reduced assimilation efficiency and reproduction | [107] |
| *Calanus helgolandicus* (Copepods) | PS | 20 μm | Reduction in carbon biomass | [108] |
| *Emys orbicularis* (Pond turtle) | PE | - | Adverse impact on the liver and kidney functioning | [109] |
| *Danio rerio* (Zebrafish) | PS | 5 and 20 μm | Inhibited liver functions and metabolism of fish | [110] |
| *Danio rerio* (Zebrafish) | PA, PE, PVC, PP | 70 μm | Damage to intestine | [116] |
| *Tripneustes gratilla* (Sea urchin) | PE | 10–45 μm | Decreased larval width and survival affected by 50% | [111] |
| *Gammarus pulex* (Amphipoda) | PET | 10–150 μm | Metabolic rate, behavior, and growth were not affected | [112] |
| *Skeletonema costatum* (Microalgae) | PVC | 1 μm and mm | Inhibition in growth and affected photosynthesis | [117] |
| *Perna viridis* (Asian green mussel) | PVC | 1–50 μm | Negative impacts on physiological functions of mussels | [113] |
| *Scrobicularia plana* (Bivalve mollusc) | PS | 20 μm | MPs inhibited antioxidant activity, damaged DNA, and caused neurotoxicity and oxidative stress. | [118] |
| *Euphausia superba* (*Antarctic Kill*) | PE | 27–32 μm | Loss in weight | [119] |
| *Oryzias lapites* (Japanese medaka fish) | LDPE | - | Resulted in formation of tumours, liver damage, and accumulation of toxic chemicals | [114] |
| *Oryzias lapites* (Japanese medaka fish) | PE | <1 mm | Adverse effects on reproduction and growth | [120] |
| *Arenicola marina* L. (Lugworms) | PVC, PS | <10 μm | Mortality and dysfunction of immune system | [115] |
| *Ostrea edulis* (Flat Oysters) | HDPE and PLA | Varying sizes | Increase in respiration rate | [121] |
| *Crangon crangon* L. (Brown shrimp) | - | 200–1000 μm | No adverse impact on the shrimp's nutritional condition | [122] |
| *Ciona intestinalis* (Sea squirt) | PS | 1 μm | Negative effects on growth and food intake | [123] |
| *Crepidula onyx* (Mollusca) | PS | 2 μm | Growth inhibition | [124] |

Note(s): PP—Polypropylene, PA—Polyamide, LDPE—Low-density polyethylene, PET—Polyethylene terephthalate, PE—Polyethylene, PVC—Polyvinyl chloride, PS—Polystyrene, PLA—Polylactic acid, HDPE—High-density polyethylene.

Green [121] studied the impact of high-density PE and PLA on *Ostrea edulis* (Flat Oysters). It was observed that the rate of respiration increased with an increase in microplastic concentration. In a study by Devriese et al. [122], it was reported that microplastic ingested by *Crangon crangon* L. (Shrimps) showed no significant negative impact on nutritional conditions. In conclusion, in-depth research is required to better comprehend the impacts of microplastics on marine biota, and further research is necessary to fill knowledge gaps.

### 4.3. Impacts of MPs on Soil

Scientists have paid minimal attention to MPs pollution in soil environments as compared to marine ecosystems, despite a more significant accumulation of MPs in terrestrial soils. Microplastic contamination is especially prevalent in agricultural and urban soils as a consequence of human activities. The soil ecosystem is exposed to a wide range of MPs due to the over-exploitation and haphazard management of plastic wastes [125]. Agricultural practices, including plowing and harvesting, influence the horizontal distribution of MPs in soil, whereas vertical distribution is governed by soil macropores and the cracking of soil [126]. The extent of transportation, deposition, and retention of MPs are influenced by numerous aspects such as (1) human activities including littering and inefficient waste handling, (2) physicochemical properties of plastics including size, density, etc., and (3) atmospheric conditions (temperature, rainfall, speed of wind). The movement of microplastics in the soil can be facilitated by a variety of plant processes such as uprooting and root development as well as various organisms (vertebrates, earthworms, etc.) inputs [127]. For instance, earthworms can swallow and excrete microplastics, the movement of anecic earthworms is capable of vertically transporting microplastics from shallow to deep soils, and geophagous earthworms are responsible for horizontally spreading them across wide areas (as shown in Figure 13) [128]. In addition, soil microarthropods have been found to consume earthworm casts containing concentrated microplastics [129]. As microplastics migrate, soil properties, including microbial diversity as well as soil structure and function are altered which could have a negative effect on plants and animals, as well as threaten the quality and safety of food.

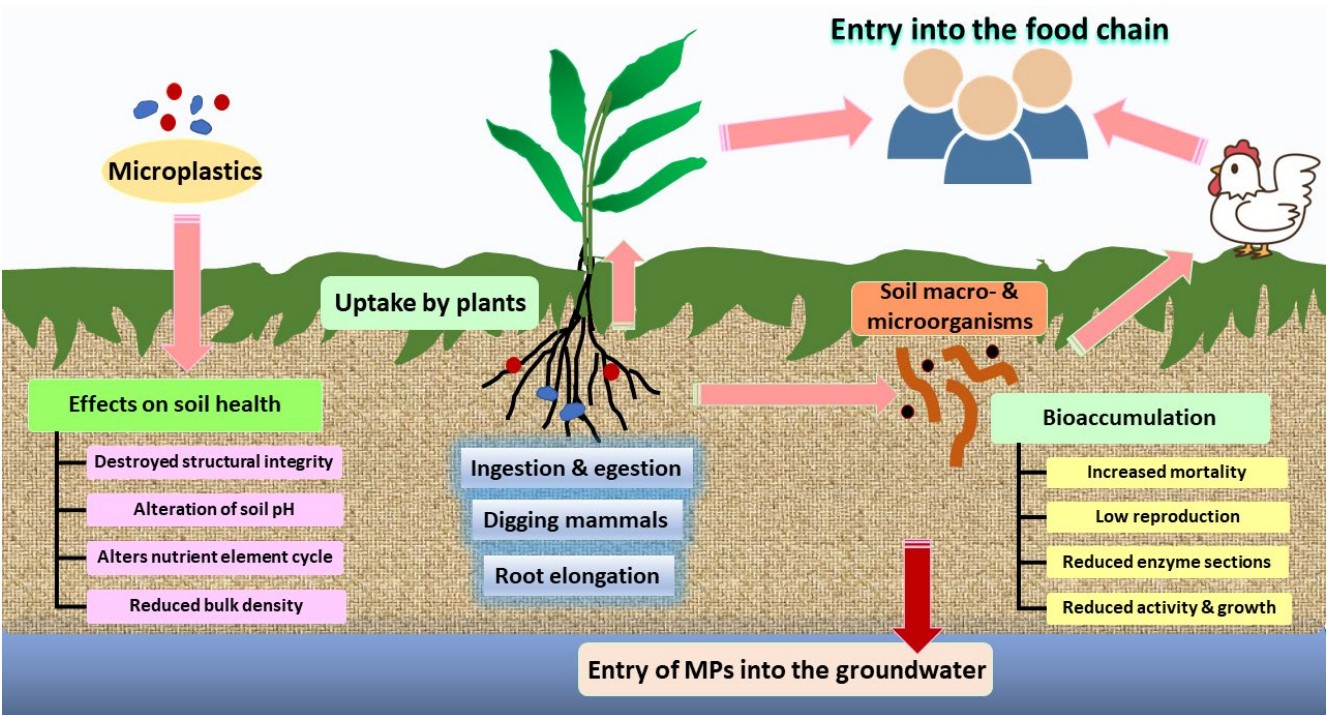

**Figure 13.** The toxic effects of microplastics on soil biota.

To study the interactions between MPs with soil biota several studies have been performed by various groups and their impacts on the health of soil biota have been examined. In contrast to aquatic habitats, the ecotoxicological impacts of MPs on soil fauna have been the subject of very little research, and most of the studies were performed in laboratories. Coa et al. (2017) studied the impact of PS (58 μm) MPs on the health of *Eisenia Foetida* (earthworms) in dry soil. They found that MPs at concentrations greater than 1% ($w/w$) decreased earthworm growth and also increased earthworm mortality [130]. Similar findings were presented by Huerta Lwanga et al. [131] where polyethylene MPs (0.2–1.2% concentration) affected the growth and mortality of *Lumbricus Terrestris* (earthworms).

Lahive et al. [132] demonstrated that varying sizes of microplastics affect the rate of reproduction of *Enchytraeus crypticus* (soil worm) differently. It was observed that smaller-sized particles (i.e., 20 μm) affected the survival and reproduction to a greater extent as compared to the larger particles (i.e., 160 μm) which could be attributed to the ingestion of a larger number of smaller-sized MPs by soil worms. A study by Rillig et al. [133] revealed that exposure of *Lumbricus terrestris* (anecic earthworm) to the PE microplastics resulted in the transport of MPs deeper into the soil. There are potential consequences of this movement including, other soil biotas that may be exposed to MPs and microplastics may remain underground for extended periods of time. A study by Lei et al. [134] stated that the exposure of PS microplastics (size 1 μm) to *Caenorhabditis Elegans* (Roundworm) for 3 days lead to reduced survival rate and growth. It was observed that nematode survival, development, and cholinergic and GABAergic neurons were most affected by the polystyrene particles. Zhu et al. [135] also demonstrated that PVC microplastic exposure to the collembolan gut leads to a 28.8 and 16.8% inhibition of reproduction and growth of the soil organisms, respectively. Song et al. demonstrated that *Achatina fulica* (snails) experienced different reductions in food intake and excretion after exposure to PET microfibers for 28 days, and microfibers caused significant villous damage to the snails' digestive tract [136]. Kim and An [137] studied the effect of PE and PS MPs on *Lobella sokamensis* (soil springtail). It was reported that MPs may accumulate in the cavities created by springtail thereby inhibiting their mobility. In a study by Ju et al. [138] a decrease in the survival and reproduction rate of *Folsomia candida* (soil springtail) was observed on exposure to different concentrations of polyethylene MPs.

Yi et al. [139] demonstrated that the impacts of MPs on the soil ecosystem are also affected by the shape of the MPs. It was observed that PP fibers were more effective at inhibiting urease and alkaline phosphatase enzyme activities than the PP microsphere [139]. Wan et al. [140] depicted that MPs alter the water evaporation of soil which may lead to the drying of soil. A significant amount of soil water evaporation occurred as a consequence of MPs' presence in the soil as they created channels for water to move through. In addition, increasing the concentrations and reducing the size of MP contributed to more pronounced effects. As a consequence, the microplastic uptake damages the key functions of soil animals which are critical to biodiversity and soil health. Details of some of these studies of MP pollution on soil and soil biota are given in Table 5.

**Table 5.** Impact of MP pollution on properties of soil and soil biota.

| Soil Biota and Properties | Polymer Types | Size | MPs Effects | References |
|---|---|---|---|---|
| *Eisenia Foetida* (Earthworm) | PS | 58 μm | Inhibition in growth and increased mortality | [130] |
| *Lumbricus Terrestris* (earthworm) | PE | ≥50 μm | Growth inhibition and mortality | [131] |
| *Lumbricus terrestris* (Earthworm) | PE | $40.7 \pm 3.8$ μm | Cellular stress | [141] |
| *Eisenia fetida* (Earthworm) | PE | 250–1000 μm | Gut damage | [142] |
| *Enchytraeus crypticus* (Soil worm) | PA | 20 and 160 μm | Rate of reproduction was affected | [132] |
| *Lumbricus terrestris* (Anecic earthworm) | PE | Varying sizes | Earthworms transported MPs deeper into the soil | [133] |

**Table 5.** *Cont.*

| Soil Biota and Properties | Polymer Types | Size | MPs Effects | References |
|---|---|---|---|---|
| *Caenorhabditis Elegans* (Roundworm) | PS | 1–5 μm | MPs caused reduction in body growth and low survival rate | [134] |
| *Caenorhabditis elegans* (Nematode) | PS | 1 μm | Oxidative stress and intestinal damage | [143] |
| *Folsomia candida* (Collembolans) | PVC | 80–250 mm | Inhibition of reproduction and growth | [135] |
| *Achatina Fulica* (snail) | PET | 76.3 μm | Reduction in food intake and damage to digestive tract | [136] |
| *Lobella sokamensis* (Soil springtail) | PE and PS | 0.47~1155 μm | Movement inhibition | [137] |
| *Folsomia candida* (Soil springtail) | PE | 281 μm | Decreased survival and reproduction rate | [138] |
| Soil enzyme (urease and phosphatase) | Membranous PE, PP microsphere and fibrous PP | - | Inhibition of enzymatic activity | [139] |
| Soil property | PE | 2, 5 and 10 mm | Increased water evaporation of soil leading to soil drying | [140] |
| *Triticum aestivum* (wheat plant) | PE | - | Inhibited the vegetative and reproductive growth | [144] |
| *Lepidium sativum* (cress seed) | - | <5 mm | Delayed germination rate and growth of its root | [145] |
| *Vicia faba* (Broad bean) | PS | 5 μm | Oxidative damage, Inhibition of plant growth, and induced genotoxicity and ecotoxicity | [146] |
| *Allium fistulosum* (Spring onions) | PEHD, PA, PES, PET, PP, and PS | Varying sizes | Affected plant performance | [147] |
| *Lactuca sativa* L. *var. ramose Hort* (Lettuce) | PS | 23 μm | Lettuce's growth rate, photosynthesis, and chlorophyll content were significantly reduced by MPs | [148] |
| *Lycopersicon esculentum Mill* (Tomato) | PET, PP, PE | 0.4–2.6 mm | MP sludge stimulated tomato plant growth but delayed the production and yield | [149] |

Note(s): PS—Polystyrene, PET—Polyethylene terephthalate, PE—Polyethylene, PP—Polypropylene, PEHD—Polyethylene high density, PA—Polyamide, PVC—Polyvinyl chloride, PES—Polyester.

In terms of microplastics' effects on terrestrial plants, there is still a lack of research and knowledge. Qi et al. [144] demonstrated that polyethylene MP films (1% *w/w*) negatively inhibited the reproductive and vegetative growth of the *Triticum aestivum* (wheat plant) in dry soil. After 8 and 24 h of MPs exposure, Bosker et al. [145] witnessed that *Lepidium sativum* (cress seed) capsules accumulated MPs and resulted in a delayed germination rate and growth of its root, respectively. Jiang et al. [146] stated that the accumulation of a large number of polystyrene MPs in the root tips of the *Vicia faba* plant could significantly result in oxidative damage, inhibit plant growth, and induced genotoxicity and ecotoxicity. Likewise, de Souza Machado et al. [147] explored the performance of *Allium fistulosum* (spring onions) when exposed to different MPs (0.2% *w/w*). The results indicated that MP exposure could alter plant biomass, elemental composition, and root traits, while the actual effects were different depending on the particle type. Gao et al. [148] demonstrated that PE microplastics demonstrated negative impacts on the growth, photosynthesis, and chlorophyll content of lettuce. Hernández-Arenas et al. [149] studied the effect of sludge containing PP, PET, and PE MPs on *Lycopersicon esculentum Mill* (Tomato). The results revealed that the growth of tomato plants in soils containing MPs was accelerated, while fruit production was delayed. As plants are an important part of the terrestrial ecosystem and MPs are prevalent, future research is needed to examine several kinds of MP particles, different soil conditions, and a wider range of plant species to investigate the potential consequences of MP pollution [150].

## 5. Protocols and Existing Infrastructure in Place for Controlling MP Release

Keeping in view the surmounting issue of microplastic accumulation in nature, international organisations and governments are endorsing concepts such as the circular economy and the six 'R's—Reduce (raw material usage), Redesign (designing reusable and recyclable products), Remove (avoid usage of single-use plastics), Reuse (refurbishment of old products), Recycle (repeated usage of products) and Recover (regeneration and

resynthesis) for sustainable growth and development [150]. Since polymers and plastics have become an indispensable part of the global economy, microplastic waste generation and release into water bodies need to be closely monitored and there is a need to undertake effective measures to minimize their detrimental impacts on the ecosystem. There is an immediate need to take action based on available evidence of MP waste while also taking precautionary approaches towards MP extraction to remove tangible future threats. Governments, organisations, industries, and the public need to work together in order to overcome these issues. Organisations such as UNEP, IMO, ICO and FAO are currently working to combat the problem of MP accumulation in aquatic bodies. An example of such action is the Canadian government which has banned the use of MPs in cosmetic products since these pellets sink to the bottom of oceans and rivers and accumulate [63]. Even after the ban, the pre-existing MPs pose a serious and imminent threat to the ecosystem since their degradation is touted to take many years.

Additionally, financial tools such as fines, taxes, fees, subsidies, deposit-refund schemes, and incentives have also proven to be effective in promoting the recycling of products which leads to a reduction in dumping and subsequent accumulation of MP pollutants in aquatic ecosystems. For instance, Ecuador requires extensive use of PET bottles for supplying clean and potable drinking water. Therefore, a bottle deposit scheme of US$ 0.02 per bottle was introduced in the country in 2011 to motivate people to deposit bottles, thereby making the collection of plastic bottles easier. It was noted that PET bottle recycling increased to 80% in 2012 from 30% in 2011, with 1.13 million bottles out of 1.40 million bottles being recycled. Measures such as the imposition of port fees (Port of Rotterdam, Netherlands), product bans (Canada), tourist fees (Galapagos Archipelago, Ecuador), and littering fines (California, USA) that involve diligence by local sand government action have also proven to be successful in tackling plastic accumulation and promotion of recycling across the world. The Indian government has enforced a country-wide ban on the use, import, manufacture, stocking, distribution, and selling of single-use plastics. Fines have also been imposed in case of failing to comply with the provisions of the ban. These steps have been taken to reduce India's contribution to global plastic waste stockpiles that have currently reached epidemic proportions.

The reduction and removal of MPs in marine ecosystems is a focal point of goal no. 14—Life Under Water (focus on marine ecosystem health) of the Sustainable Development Goals put forth by the United Nations. Particularly, it aims to improve aquatic ecosystem health by reducing sources of marine pollution, especially microplastic release by 2025. In the year 2019, member countries of the G20 summit released a statement termed the *Osaka Blue Ocean Vision* wherein the Ministry of Foreign Affairs of Japan launched the MARINE initiative that aims to reduce MP pollution of oceans and assist developing countries in plastic waste management by using the life cycle approach and innovative solutions [151]. The US Marine Debris Program developed the NOAA marine debris program that aims at the removal of MP from marine bodies without the requirement of any sophisticated instruments [152]. Collection and analysis systems such as the Manta net, Albatross device, and PLEX (PLastic EXplorer) instrument have also been involved for water remediation purposes [153–155].

In recent years, attempts have been made to find materials that can serve as alternatives to commodity plastics. The use of compostable and biodegradable plastics and paper is being promoted for use as packaging material while natural products such as walnut shell powder and mineral powder have replaced microsized plastic pellets in cosmetic products. Polylactic acid (PLA), and starch, sugarcane, and mushroom-based biomaterials are being promoted for use in various applications as substitutes for PE and PS due to their as benign nature and biodegradability. Although these substitutes are biocompatible and degrade more easily than microplastics, their use is not always economical and environment-friendly since they raise production costs and require more exploitation of natural resources. Hence, there is still scope for improvement in the adoption of these alternative materials. Efforts are being made at multiple levels to overcome the problem of MP pollution but despite

all these efforts, the influx of MPs into water bodies, soil, and human, plant, and animal physiology is rising at an astonishing rate. Hence, there is a need for strict rules and stringent action from the policymakers and the public to safeguard the future.

## 6. Conclusions

Global plastic production and usage are expected to increase in the coming years, which may lead to increased microplastic pollution in the environment. In the current research, it has been determined that microplastics have diverse sources, and the way in which they occur, transport, and evolve depends on a wide range of natural conditions as well as their physicochemical characteristics (e.g., size, crystallinity, shape, density, etc.). The quantification and identification of microplastics are essential prerequisites to ensure their efficacious removal. The combined use of visual analysis and spectroscopic techniques is especially helpful in the detection of microplastics in aquatic systems but they possess certain drawbacks. Thus, keeping in view the extensive amount of MP waste generated globally, efforts need to be made to develop better techniques for MP detection and identification to minimize misidentification. To extract MP waste from wastewater, many treatment methods have been developed to facilitate rapid and efficient removal. Conventional and advanced methods such as CFS, membrane filtration, adsorption, and biological degradation are often used in conjunction and have the potential to exhibit removal efficiencies as high as >99%. Although, issues such as membrane fouling, adsorption site blockage, particle selectivity, and lack of reusability plague these treatment methods, and the effect of size, shape, and polymer type on the removal efficacy is not very clear. These limitations call for further research into these methods to improve the MP removal capability of these technologies. Most of the studies have examined the toxicology of microplastics in the marine environment, but their impact on soil biota and human health has not been fully explored. Moreover, further research should be conducted on how microplastics affect human cells in vivo. Since microplastic waste accumulation has reached epidemic proportions at the global scale, certain protocols and infrastructure has been enforced to reduce further generation MPs and attempts are being made to combat MP waste accumulation at the international, national, and local levels.

**Author Contributions:** The presented work was carried out by the contribution of all the authors. Conceptualisation: B.P. and J.P.: Writing—original draft: B.P. and J.P.: Visualisation: P.S., R.K. and A.K.: Review and editing: P.S., R.K., T.K.T., S.K. and A.K. All authors have read and agreed to the published version of the manuscript.

**Funding:** This research received no external funding.

**Institutional Review Board Statement:** Not applicable.

**Data Availability Statement:** Not applicable.

**Acknowledgments:** The authors would like to thank Delhi Technological University, DTU (India) for providing the necessary infrastructure to carry out the research. Author BP would also like to thank CSIR, India for providing SRF fellowship.

**Conflicts of Interest:** The authors declare no conflict of interest.

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
