# Peer review of "Microplastics in the Ecosystem: An Overview on Detection, Removal, Toxicity Assessment, and Control Release"

_water, doi:10.3390/w15010051_

Round 1

Reviewer 1 Report

The manuscript entitled "Microplastics in the ecosystem: An overview on detection, removal, toxicity assessment, and control release” presents a significant work. But manuscript requires major revision on basis of following points:

-The title seems good.

-The abstract seems to be good. Please add one more introductory line of your objective in beginning of abstract.

-Please explain the acronyms: SEM-EDX, PLM, FTIR, Raman, TG-DSC, and GC-MS

-Research gap should be delivered on more clear way with directed necessity for the future research work.

-The novelty of the work must be clearly addressed and discussed, compare previous research with existing research findings and highlight novelty.   -There is no explanation of the results: add SEM-EDX figures, FTIR spectra, peaks, and GC chromatograms...

-For duplication: We checked and found that there are still some duplication in your paper. Please find the duplication report

Author Response

The manuscript entitled "Microplastics in the ecosystem: An overview on detection, removal, toxicity assessment, and control release” presents a significant work. But manuscript requires major revision on basis of following points:

Reply: We are highly thankful for the reviewers for constructive comments to improve the quality of the manuscript 

-The title seems good.

Reply: thanks 

-The abstract seems to be good. Please add one more introductory line of your objective in beginning of abstract.

Reply:  Thanks, as suggested abstract has been revised carefully

-Please explain the acronyms: SEM-EDX, PLM, FTIR, Raman, TG-DSC, and GC-MS

Reply: Noted and done 

-Research gap should be delivered on more clear way with directed necessity for the future research work.

Reply: thanks for your suggestions, we tried our best to revise the manuscript critically by taking care of other reviewers too

-The novelty of the work must be clearly addressed and discussed, compare previous research with existing research findings and highlight novelty.   -There is no explanation of the results: add SEM-EDX figures, FTIR spectra, peaks, and GC chromatograms...

Reply: thanks for your suggestions, manuscript has been critically revised, kindly see the highlighted tables and text

-For duplication: We checked and found that there are still some duplication in your paper. Please find the duplication report

Reply: thanks, necessary amendment has been done thoroughly 

Reviewer 2 Report

The desirable properties and benefits of plastics over other materials enable them to deliver undeniable benefits to humanity. However, the continued production and dependence on plastics has led to unprecedented environmental pollution. In particular, microplastics (MPs) is now listed as one of the most threatening group of Contaminants of Emerging Concern (CEC) because (i) their toxicity to humans, aquatic organisms and ecosystems is not fully understood, (ii) they have the potential to concentrate other pollutants (including bacteria, heavy metals, viruses) on their surfaces, thus acting as a vector that can amplify the toxicity of such toxicants to fauna and flora, (iii) there are no universally agreed sampling protocols and analytical techniques for investigation of MPs. For this reason, the review ‘‘Microplastics in the ecosystem: An overview on detection, removal, toxicity assessment, and control release’’ is on a topic of both environmental and public relevance, that warrant publication. It is incontestable that various detailed reviews on MPs have been published between 2020 and 2022 (see for example, Ateia et al. 2022; Hampton et al. 2022; Papp et al., 2022; Yang et al., 2022). Against this background, a further review on this topic will definitely need to emphasize nascent advances, findings and/or come to further conclusions and highlight the salient directions for future research. The current submission partly fulfils these criteria, as it tends to mix up ideas on the topic. I therefore recommend that the manuscript be revised based on the following comments to improve it to a standard that is publishable in a journal with international audience.

Cited literature

Ateia et al. (2022). Emerging investigator series: microplastic sources, fate, toxicity, detection, and interactions with micropollutants in aquatic ecosystems – a review of reviews. Environmental Science: Processes Impacts, 24, 172-195.

Yang et al. (2022). A comparative review of microplastics in lake systems from different countries and regions Chemosphere, 286, 131806.

Hampton et al. (2022). Characterizing microplastic hazards: which concentration metrics and particle characteristics are most informative for understanding toxicity in aquatic organisms? Microplastics and Nanoplastics, 2, 20.

Papp et al. (2022). Origin, environmental presence and health effects of microplastics. Acta Biologica Szegediensis, 66(1), 75-84.

1. Title

I found some issues with the title, (i) this draft may not in any way be considered an ‘‘overview’’ as the authors stated, (ii) detection to me seems like no full quantification is involved, and yet the latter is the most important as MPs are considered to be universally ubiquitous, (ii) toxicity assessment could come second before removal strategies. For these reasons, the title of this draft could logically be revised to read: Microplastics: A review on environmental levels, toxicity assessment, removal and control strategies.

Main text

L107-108: This sentence is rather confusing. Of course, primary MPs are industrially produced in miniaturized sizes, usually as fragments or pellets e.g., industrial nurdles, microbeads in cosmetics and microfibers in apparels. Check Ateia et al. (2022) above for further details.

L127-129: In Fig. 2, remove the heading as the caption is also there. The figure should be substantially improved; the various processes indicated should actually act on the macroplastics to produce MPs. The current figure shows that these processes are themselves source of MPs. Also, the caption could be revised to: Fig. 2. Mechanisms of production of secondary MPs from macroplastics.

L186-260, L261-288: Various discussions are made but with no single reference. Since this is a review, it is unexpected that there is any original contribution from the authors. Please cite the sources of these information.

L260: In Fig. 4, hot needle test is also used for ruling out false positives.

It should also be pointed out which challenges are associated with each technique/method, and why there is lack of universal methods for MPs analyses. What challenges are associated with MPs quantification (e.g., background & cross contamination, false positives, challenges with sampling protocols which limits comparison among studies, etc)

L261: This categorization of the removal strategies needs to be revisited. There are;

(i)  Chemical methods: such as chlorination, coagulation, flocculation, agglomeration, use of nanocomposites.

(ii)    Physical methods; basically adsorption, sedimentation and filtration (ultrafiltration).

(iii)  Biological methods; utilizing enzymes, activated sludge, hyperthermophilic composting, biofilters and active ingestion. 

Literature

Ahmed et al. Critical review of microplastics removal from the environment. Chemosphere 2020, 293, 133557.

Lapointe et al. (2020). Understanding and improving microplastic removal during water treatment: impact of coagulation and flocculation. Environmental Science and Technology, 54(14), 8719–8727.

Arossa et alMicroplastic removal by Red Sea giant clam (Tridacna maxima). Environ. Pollut. 2019, 252, 1257–1266.

Auta et al. Screening of Bacillus strains isolated from mangrove ecosystems in Peninsular Malaysia for microplastic degradation. Environ. Pollut. 2017, 231, 1552–1559.

Changliang et al. Experimental study on removal of microplastics from aqueous solution by magnetic force effect on the magnetic sepiolite. Separat. Pur. Technol. 2022, 288, 120564.

Chen et al. Enhanced in situ biodegradation of microplastics in sewage sludge using hyperthermophilic composting technology. J. Harzar. Mat. 2020, 384, 121271.

Liu et al. Microplastics removal from treated wastewater by a biofilter. Water, 2020, 12, 1085.

Liu et al. Transfer and fate of microplastics during the conventional activated sludge process in one wastewater treatment plant of China. Chem. Eng. J.  2019, 362, 176–182.

Maity et al. Emerging Roles of PETase and MHETase in the Biodegradation of Plastic Wastes. Appl. Biochem. Biotechnol. 2021193, 2699–2716.

Shi et al. (2022). Removal of microplastics from water by magnetic nano-Fe3O4. Sci. Tot. Environ. 802, 149838.

Tang et al. Removal of microplastics from aqueous solutions by magnetic carbon nanotubes. Chem. Eng. J. 2021, 406, 126804.

Zhang et al. Boron-doped carbon nanoparticles for identification and tracing of microplastics in “Turn-on” fluorescence mode. Chem. Eng. J. 2022, 435, 135075.

Zhou et al. Removal of polystyrene nanoplastics from water by CuNi carbon material: The role of adsorption. Sci. Total Environ. 2022, 820, 153190.

-The authors should cite out the challenges with each of these methods, and suggest which of them are the most feasible based on resources required, removal efficacy etc.

L703: There are also issues surrounding use of biodegradable plastics. Use of biodegradable plastics is apparently a limited part of the technical solution, as their biodegradation require certain conditions. Moreover, the formation of ‘‘bio-microplastics’’ cannot be ruled out (Choe et al., 2021).

Choe et al. Bridging three gaps in biodegradable plastics: miconceptions and truths about biodegradation. Front. Chem. 20219, 671750.

Author Response

Thanks for your comments and suggestions to improve the quality of the manuscript critically. In light of all comments, we have revised the manuscript and incorporate all suggestions, kindly see the highlighted text and table for your positive view and decision. Besides, suggested research and review articles was very supportive during revision. 

Once again thanks 

Round 2

Reviewer 1 Report

The authors did much work and they have improved the quality of the document. However, there are still some duplication in your paper.  

Please find the duplication report.

Author Response

Dear Authors, 

As per your suggestions, manuscript has been revised carefully and plagiarism has been checked through Turnitin software (13%). 

Thanks and Regards  

Reviewer 2 Report

Accept

Author Response

Thanks for your acceptance, English has been checked carefully and revised